# Distinct temporal integration of noradrenaline signaling by astrocytic second messengers during vigilance

Yuki Oe[1]*, Xiaowen Wang[1,2], Tommaso Patriarchi[3,4,5], Ayumu Konno [6,7], Katsuya Ozawa[1], Kazuko Yahagi[1], Hirokazu Hirai [6,7], Lin Tian [3], Thomas J. McHugh [1] & Hajime Hirase [1,2,8]*

Astrocytes may function as mediators of the impact of noradrenaline on neuronal function. Activation of glial α1-adrenergic receptors triggers rapid astrocytic $Ca^{2+}$ elevation and facilitates synaptic plasticity, while activation of β-adrenergic receptors elevates cAMP levels and modulates memory consolidation. However, the dynamics of these processes in behaving mice remain unexplored, as do the interactions between the distinct second messenger pathways. Here we simultaneously monitored astrocytic $Ca^{2+}$ and cAMP and demonstrate that astrocytic second messengers are regulated in a temporally distinct manner. In behaving mice, we found that while an abrupt facial air puff triggered transient increases in noradrenaline release and large cytosolic astrocytic $Ca^{2+}$ elevations, cAMP changes were not detectable. By contrast, repeated aversive stimuli that lead to prolonged periods of vigilance were accompanied by robust noradrenergic axonal activity and gradual sustained cAMP increases. Our findings suggest distinct astrocytic signaling pathways can integrate noradrenergic activity during vigilance states to mediate distinct functions supporting memory.

[1] RIKEN Center for Brain Science, Wako, Saitama, Japan. [2] Center for Translational Neuromedicine, Faculty of Medical and Health Sciences, University of Copenhagen, Copenhagen, Denmark. [3] Department of Biochemistry and Molecular Medicine, University of California, Davis, CA, USA. [4] Institute of Pharmacology and Toxicology, University of Zurich, Zurich, Switzerland. [5] Neuroscience Center Zurich, University of Zurich, Zurich, Switzerland. [6] Viral Vector Core, Gunma University Initiative for Advanced Research, Maebashi, Gunma 371-8511, Japan. [7] Department of Neurophysiology & Neural Repair, Gunma University Graduate School of Medicine, Maebashi, Gunma, 371-8511, Japan. [8] Brain and Body System Science Institute, Saitama University, Saitama, Japan. *email: oe@brain.riken.jp; hirase@sund.ku.dk

Astrocytes play multiple roles in neural circuit dynamics by modulating synaptic transmission and plasticity, influencing energy availability and shaping the extracellular environment. Ample evidence had implicated astrocytic $Ca^{2+}$ elevations, generated via G protein-coupled receptor (GPCR) metabotropic pathways, in signaling to neurons[1–3]. For example, GPCRs coupled to the Gq signaling pathway (Gq-GPCR) activate phospholipase C, which in turn produces $IP_3$, triggering $IP_3$ receptors on the endoplasmic reticulum to initiate internal $Ca^{2+}$ release.

Astrocytic GPCRs include receptors for volume-transmitted subcortical neuromodulators such as noradrenaline (NA) and acetylcholine, which can trigger $Ca^{2+}$ elevations in astrocytes in vivo[3,4], promoting synaptic plasticity[4–7]. In awake mice, startle causes cortical astrocytic $Ca^{2+}$ elevations by the activation of the Gq-coupled α1-adrenergic receptor (α1-ARs)[8,9]. While the role of these astrocytic $Ca^{2+}$ elevations in learning and memory remains debated[10,11], a recent study has reported that chemogenetic activation of Gq-GPCRs in hippocampal astrocytes, but not in neurons, enhances context fear memory[12]. Astrocytes also express Gs-coupled β-ARs which increase intracellular cAMP levels. Among many β-AR-dependent intracellular processes, glycogenolysis and the resultant lactate shuttle have been shown crucial to establish memory in rats[13–15]. However, elevated expression of the adenosine $A_{2A}$ adenosine receptor, which is also a Gs-GPCR, occurs in Alzheimer's disease patients, and sustained astrocytic activation of Gs signaling by chemogenetics in mice has been demonstrated to impair memory consolidation[16]. Thus, a clearer understanding of how these pathways are activated during learning is required.

Noradrenergic (NAergic) innervations to the cortex originate from neurons in the locus coeruleus (LC)[17,18]. LC neurons fire at low basal rates (1–3 Hz) at rest and display phasic burst firing during attentive and vigilant states in awake animals[19]. It has been widely acknowledged that NAergic activity is involved in the formation of fear memory, with deficit resulting from chemical lesion of the LC[20] or pharmacological blockade of ARs[21–23], and genetic manipulation of NA release have been shown to lead to significant changes in a variety of other memory tasks[24–26]. However, despite the well-documented relationship between NA and memory formation, how the distinct firing patterns of NA neurons differentially impact astrocytic function is not known, particularly in terms of the role of astrocytic cAMP signaling.

Here, we sought to investigate how NAergic activity modulates astrocytic $Ca^{2+}$ and cAMP dynamics during fear conditioning. To this end, we utilized a recently developed Epac1-based red fluorescent cAMP probe, Pink Flamindo[27], that can be imaged in conjunction with green GCaMP $Ca^{2+}$ probes[28,29]. First, we employed optogenetic stimulation of LC/NA axons in the cortex to characterize how synaptically released NA drive these two distinct astrocytic second messengers in vivo. Next, we demonstrate that astrocytic cAMP levels depend on the level of vigilance, with increases associated with prolonged phasic activity of LC/NA axons. Together these experiments demonstrate distinct NA signaling modes of astrocytes during learning.

## Results

To express fluorescent probes selectively in NAergic neurons, we injected Cre-dependent adeno-associated viral (AAV) vectors expressing EYFP or ChR2-EYFP into the LC of noradrenaline transporter (NAT)-cre mice[30] (Fig. 1a). Three weeks after AAV injection, EYFP was expressed in tyrosine hydroxylase (TH, an essential enzyme required for the synthesis of NA and dopamine)-positive cells in the LC, but not in the ventral tegmental area (Fig. 1b, c, f, Supplementary Fig. 1). LC-originated NAergic

axons were visualized in the cerebral cortex (Fig. 1d). Following unilateral injection of AAV-DJ/8-EF1a-DIO-EYFP, ~50% of TH+ fibers exhibited visible expression of EYFP in the auditory cortex (Fig. 1e, g), indicating efficient and specific expression of the fluorescent probe. These results are in agreement with the known predominant NAergic projections from LC.

**NA levels activate distinct astrocytic $Ca^{2+}$ and cAMP surges.** To achieve $Ca^{2+}$ and cAMP 2-photon imaging in astrocytes, we expressed GCaMP7 or Pink Flamindo in the parietal cortex via AAVs with the astrocyte-specific GFAP promoter (Supplementary Fig. 2). First, we sought to characterize astrocytic $Ca^{2+}$ activity in response to different temporal patterns of NAergic axon photostimulation (PS, see Methods). Two-photon imaging of GCaMP7 signals found astrocytic $Ca^{2+}$ increases following a 2-s PS (146.5 ± 6.5%), with the magnitude increasing with PS duration, with PS over 3 s nearly saturating GCaMP7 peak amplitude ($\Delta F/F$ 222.6 ± 13.4%). A vast majority of astrocytes (84.4 ± 4.3%) showed $Ca^{2+}$ elevations with 5-s PS ($\Delta F/F$ 244.6 ± 9.6%) (Supplementary Movie 1), while a 1-s PS did not evoke a detectable $Ca^{2+}$ increase ($\Delta F/F$ 102.0 ± 1.3%) (Fig. 1h, j, l, n, o). By comparison, Pink Flamindo responses reported no significant change in cAMP levels following 3- and 5-s PS (100.9 ± 0.5% and 103.7 ± 1.4%, respectively). However, PS longer than 10 s elicited significant cAMP increases ($\Delta F/F$ 106.7 ± 0.6%), with a 30-s PS ($\Delta F/F$ 115.0 ± 0.9%) elevating cAMP in a majority of astrocytes (90.0 ± 4.0%) (Fig. 1i, k, m, p, q) (Supplementary Movie 2).

We next investigated which adrenoreceptor subtype is responsible for astrocytic $Ca^{2+}$ and cAMP elevations by pharmacological blockade (Fig. 1r–t). As expected, we found that $Ca^{2+}$ elevations were blocked by the α-1 receptor antagonist prazosin but not by the non-selective β receptor antagonist propranolol (GCaMP $\Delta F/F$, prazosin: before = 310.4 ± 33.3% vs. after = 117.5 ± 17.9%; propranolol: 316.4 ± 17.8% vs. 322.6 ± 24.5%). By contrast, cAMP elevations were not affected by prazosin but blocked by propranolol (Pink Flamindo $\Delta F/F$, prazosin 113.9 ± 3.4% vs. 111.7 ± 2.4%; propranolol: before = 115.8 ± 2.2% vs. after = 102.5 ± 0.7%). We further examined the contributions of β-1 and β-2 receptors using the respective antagonists betaxolol and ICI 118,551. We found a dominant contribution of β-1 activation for astrocytic cAMP (Fig. 1u, Betaxolol: 114.1 ± 2.6% vs. 102.9 ± 1.2%, ICI 118,551: 114.5 ± 3.1% vs. 110.2 ± 2.9%), which is consistent with published brain cell type RNA-sequencing transcriptome databases[31,32].

To gain insight into how these second messenger responses reflect the dynamics of synaptically released NA in the extracellular space, we utilized nLight, a recently developed genetically encoded fluorescent probe for extracellular NA[33] (Supplementary Fig. 3). As before, ChR2 was expressed specifically in LC NAergic neurons and nLight was expressed in neurons in the parietal cortex by AAV9-hSynI-nLight. As nLight is a membrane protein, it showed a quasi-uniform pattern in the neuropil at the mesoscopic scale (Fig. 2a).

We first verified nLight NA sensitivity following varying PS durations. The results in Fig. 1 showed that while 1-s PS did not cause visible astrocytic responses, 3–5-s PS induces astrocytic $Ca^{2+}$ elevations, and ≥10-s PS triggered both $Ca^{2+}$ and cAMP increases, thus we stimulated NA fibers accordingly. nLight responses were observed with PS ≥ 3 s and became larger with longer PS (Fig. 2b, c, Supplementary Movie 3; 1 s: 100.7 ± 0.4%, 3 s: 101.4 ± 0.2%, 5 s: 102.3 ± 0.4%, 10 s: 104.4 ± 0.9%, 30 s: 106.9 ± 1.9%). These results support our conclusion that astrocytic $Ca^{2+}$ and cAMP elevations driven by PS were caused by NA, and suggests that astrocytic $Ca^{2+}$ can be elicited by lower extracellular NA levels, while astrocytic cAMP requires higher

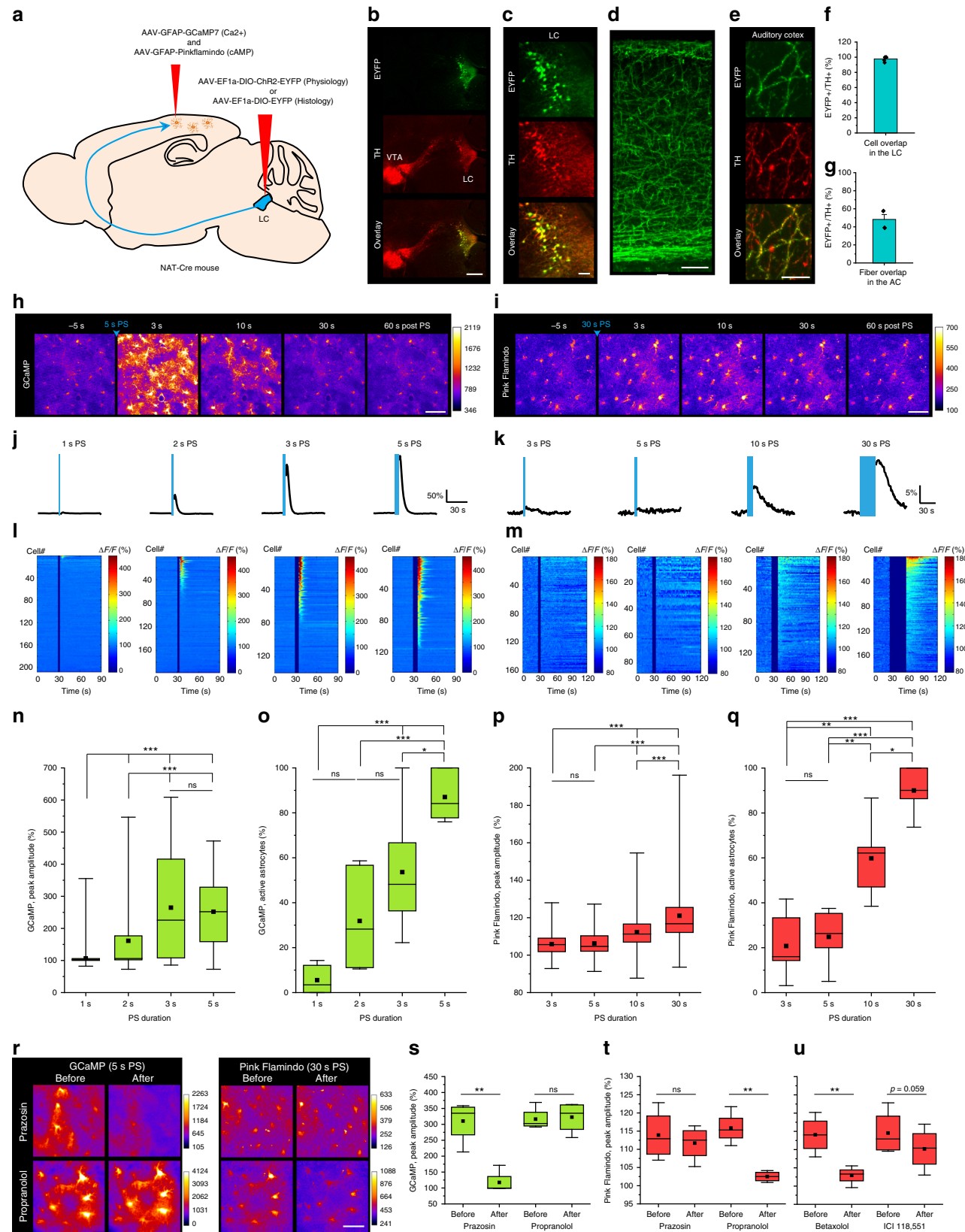

extracellular NA levels. To confirm selectivity of nLight to NA, we examined PS-triggered nLight responses in the presence of desipramine, a selective NA reuptake inhibitor. Desipramine dramatically prolonged the nLight response, confirming that nLight properly detects NA (Fig. 2d, e; 30 s after PS:

before = 107.8 ± 1.9%, after = 106.6 ± 1.5%; 300 s after PS: before = 97.7 ± 1.5%, after = 104.0 ± 0.8%).

**Temporally distinct astrocytic Ca$^{2+}$ and cAMP surges by NA.** To understand the coordination of Ca$^{2+}$ and cAMP signaling, we

**Fig. 1 Ca$^{2+}$ and cAMP responses of cortical astrocytes to noradrenergic afferent activation. a** Illustration of virus injections. This image was adapted from Allen Mouse Brain Atlas. **b** EYFP in LC neurons 3 weeks after AAV delivery (green). LC is visualized by tyrosine hydroxylase (TH) immunohistochemistry (red). VTA: ventral tegmental area. **c** Magnified images of LC. **d** Innervation of LC NAergic neurons in the cerebral cortex. **e** EYFP-labeled LC-neuronal NAergic fibers overlap with TH+ fibers in the cortex. **f** Virtually all TH+ cells express EYFP in the LC ($n = 5$ images, 5 mice). **g** Unilateral AAV microinjection to the LC labels 50% of TH+ fibers in the cortex ($n = 4$ images, 3 mice). **h, i** Astrocytic Ca$^{2+}$ and cAMP responses to optogenetically activated NAergic axons in the cortex with 5-s PS (**h**) and 30-s PS (**i**). **j, k** Average Ca$^{2+}$ and cAMP responses of cortical astrocytes to varying NAergic fiber photostimulation (PS) length. **l, m** Individual cell responses are plotted and sorted by response amplitude. **n, o** Astrocytic Ca$^{2+}$ activity analyzed by peak amplitude (**n**) ($n = 208, 194, 139$ cells, 144 cells) and active astrocytes (**o**) ($n = 6$-7 sessions, 5 mice). **p, q** Astrocytic cAMP activity analyzed by peak amplitude (**p**) ($n = 161, 87, 142, 186$ cells) and active astrocytes (**q**) ($n = 5$-6 sessions, 6 mice). **r-u** Pharmacological dissection of LC/NA axon-evoked astrocytic Ca$^{2+}$ and cAMP. Representative images of GCaMP and Pink Flamindo with prazosin or propranolol application (**r**). Comparison of GCaMP (**s**) and Pink Flamindo (**t**) signals after prazosin or propranolol application. Pink Flamindo responses after betaxolol or ICI 118,551 application (**u**) ($n = 4$ mice for all plots). Analysis was performed from individual cells for **n** and **p** and from averages of individual sessions for the others. All astrocyte responses represent somatic signals hereinafter unless otherwise noted. Scale bars: **b**, 500 μm; **c, d, h, i**, 100 μm; **r**, 50 μm; **e**, 10 μm. Bar graphs: mean + SEM. Box plots: box range, 25–50–75% quatile; square, mean; whiskers, max–min. One-way ANOVA with Turkey's test: **n-q**; paired $t$-test: **s-u**; *$p < 0.05$, **$p < 0.01$, ***$p < 0.001$.

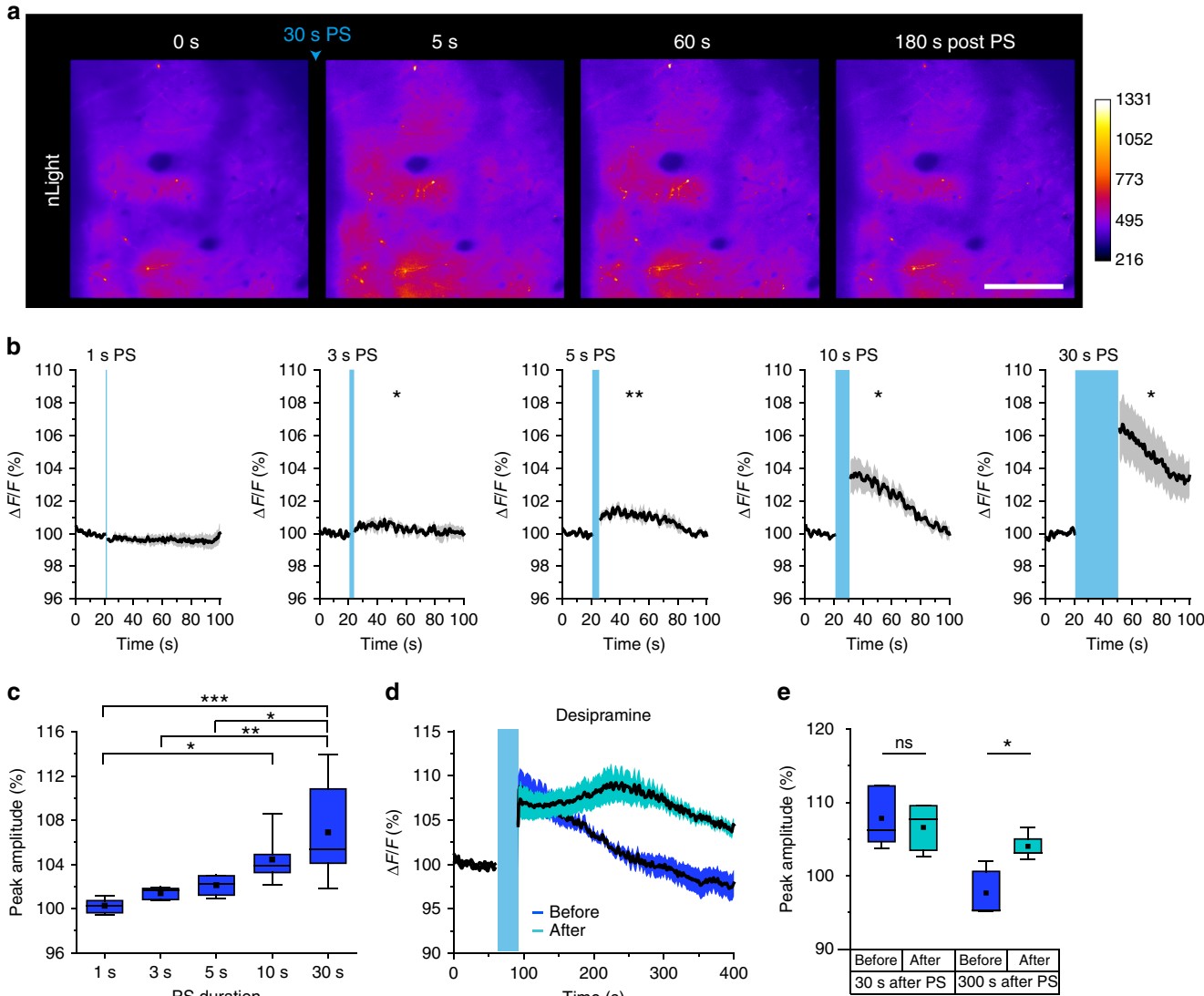

**Fig. 2 LC/NA-driving PS duration is highly correlated with extracellular NA concentration in the cerebral cortex. a** Representative cortical images of nLight before and after 30-s LA/NA axon PS. Scale bar: 100 μm. **b** nLight responses by varying PS duration. Asterisks indicate significant differences of peak amplitude after PS compared with the basal signal ($n = 5$-6 sessions, 4 mice). **c** Comparison of peak nLight response amplitude for the data in **b** ($n = 5$-6 sessions, 4 mice). **d, e** nLight signals before and after desipramine injection (NA reuptake inhibitor, 10 mg/kg, i.p. 30 min before imaging). Traces represent mean ± SEM (**d**). Peak amplitude was quantified at 30 and 300 s after PS (**e**, $n = 5$ sessions, 4 mice). Analysis was performed from the average of individual sessions. Box plots: box range, 25–50–75% quatile; square, mean; whiskers, max–min. One-way ANOVA with Turkey's test: **c**; paired $t$-test: **b**, **d**; *$p < 0.05$, **$p < 0.01$, ***$p < 0.001$.

next imaged GCaMP and Pink Flamindo simultaneously. Cortical astrocytes were imaged with 1040–1060 nm wavelength laser which while optimized towards Pink Flamindo permits the observation of GCaMP responses with a comparable dynamic range. We employed two patterns of PS, which provided equal stimulation duration but with distinct temporal patterns. In the "repetitive" stimulation regime, 3-s PS, which induced $Ca^{2+}$ elevations but not cAMP elevations in astrocytes, was repeated $10\times$ with 7 s inter-stimulus intervals, whereas in the "single" continuous stimulation regime, 30 s of continuous PS was applied. The repetitive stimulation induced significant $Ca^{2+}$ elevations in astrocytes ($\Delta F/F$ 120.0 ± 6.6%) as well as a small elevation of cAMP (104.4 ± 1.6%) (Fig. 3a, c). By contrast, the continuous stimulation reliably and robustly induced both $Ca^{2+}$ and cAMP elevations (Fig. 3b, c) ($\Delta F/F$ GCaMP: 115.7 ± 4.0%, Pink Flamindo: 117.7 ± 4.8%). These results suggest that the two second messengers have distinct dynamics in that (1) the activation threshold is lower for $Ca^{2+}$ than cAMP (Fig. 1n, o, p, q), (2) their dynamics depend on NAergic activity patterns (Fig. 3a–c), and (3) the duration of the cAMP signal is longer than that of $Ca^{2+}$ (Fig. 3b, d) (response duration: GCaMP: 17.0 ± 2.3 s, Pink Flamindo: 38.0 ± 7.3 s).

To gain more insight into the differential time course of $Ca^{2+}$ and cAMP dynamics, we examined $Ca^{2+}$ and cAMP responses during longer PS (120 s), mimicking a saturated extracellular NA environment. To accommodate two-photon image acquisition we introduced brief photostimulus-free periods during the PS (Fig. 3e, see Methods). Under these conditions, we obtained distinct time courses for $Ca^{2+}$ and cAMP signals during activation of NAergic fibers. $Ca^{2+}$ activity showed an immediate increase after the start of PS (time to peak: 11.3 ± 1.3 s) and gradually decreased to basal levels after 70 s despite the presence of additional PS. $Ca^{2+}$ increase was not observed thereafter (Fig. 3f, g, j). On the other hand, cAMP signals took 30–40 s (32.9 ± 4.2 s) to reach the peak and did not return to basal level during the PS (Fig. 3f, h, j). Such slow dynamics of cAMP cannot be explained by the characteristics of Pink Flamindo which reaches a peak within 5 s even at low cAMP concentrations[27]. Therefore, these results indicate innate properties of individual astrocytic second messengers. Regarding duration, cAMP (118.5 ± 13.2 s) was significantly longer than $Ca^{2+}$ (66.3 ± 5.3 s) (Fig. 3k), suggesting longer-lasting effect of cAMP than $Ca^{2+}$ in astrocytes. Notably, extracellular NA had similarly longer dynamics (Fig. 3i–k) (time to peak: 67.3 ± 4.7 s, duration: 153.1 ± 23.1 s) than astrocytic cAMP, which suggests that a long-lasting effect of astrocytic cAMP can take place under sustained high extracellular NA levels, while astrocytic $Ca^{2+}$ response is sensitive only to the initial extracellular NA increase.

Next, we took advantage of simultaneous $Ca^{2+}$ and cAMP measurements to ask if $Ca^{2+}$ and cAMP activities are correlated in individual astrocytes. The amplitude of signal changes after 30-s PS were divided into responsive ("high") or less sensitive cells ("low") at the median for their $Ca^{2+}$ and cAMP responses (Fig. 3l). Interestingly, while 70% of the astrocytes showed similar response tendencies for both $Ca^{2+}$ and cAMP, the remaining population exhibited preferential responses in either $Ca^{2+}$ or cAMP (Fig. 3m). This heterogeneity of these responses to NAergic activation was visualized by plotting $\Delta Ca^{2+}$ vs. $\Delta cAMP$ for individual astrocytes (Fig. 3n, $R^2 = 0.321$, slope of linear regression = 0.557). These results suggest two possibilities. First, the expression ratio of each adrenoreceptor subtypes may not be the same among individual astrocytes, leading to heterogeneity in astrocytic responses to NA. Second, expression levels of the fluorescent probes could differ across cells due to non-uniform viral gene transfer, generating the apparent heterogeneity. To assess the latter, we examined basal GCaMP intensity vs. basal

Pink Flamindo intensity and found them highly correlated. Moreover, neither basal GCaMP intensity nor basal Pink Flamindo intensity correlated to relative signal amplitude (Supplementary Fig. 4). These two observations argue against differential probe expression as the cause of NA response heterogeneity and support the idea that the proportion of adrenoreceptor subtypes may underlie the heterogeneity.

**Bursting NAergic activity induces astrocytic $Ca^{2+}$ surges.** To understand the mechanism behind the temporally distinct astrocytic second messenger dynamics, we imaged cortical NAergic axonal activities with a faster responsive $Ca^{2+}$ probe GCaMP6.f (Fig. 4a, Supplementary Fig. 5). In awake conditions, cortical NAergic axons exhibited continual $Ca^{2+}$ activities in awake mice in a range of approximately 0.5–0.6 Hz (Fig. 4b, d, f) (0.56 ± 0.04 Hz), whereas such activities were not observable under deep isoflurane anesthesia (Supplementary Fig. 6). To reveal how this NAergic activity influences astrocytes, we simultaneously imaged astrocytic and NAergic axonal $Ca^{2+}$ activities with R-CaMP1.07 and GCaMP6.f, respectively. We observed two types of NAergic $Ca^{2+}$ activity patterns: single peak (SP) signals whereby $\Delta F/F$ returns to the basal level before next firing initiation and multipeak (MP) signals whereby $Ca^{2+}$ events occur before reaching to the base level forming multiplet bursts. Interestingly, astrocytic $Ca^{2+}$ elevations occur reliably with MP signals (Fig. 4d, e, g, Supplementary Movie 4; coincidence: 51.4 ± 5.5%), particularly those with longer durations (Fig. 4h, i; duration: 4.95 ± 0.32 s with $Ca^{2+}$, 3.26 ± 0.20 s without $Ca^{2+}$). The histogram of individual MP signal durations showed that the mode was 2–3 s in the absence of an astrocytic $Ca^{2+}$ surge, but 4–5 s in the presence of an astrocytic $Ca^{2+}$ surge. Furthermore, MP signals longer than 6 s always co-occurred with an astrocytic $Ca^{2+}$ surge, consistent with the optogenetic experiments in Fig. 1, while short duration SP signals (duration: 1.72 ± 0.05 s) never resulted in detectable astrocytic $Ca^{2+}$ elevation (Fig. 4j, Supplementary Movie 5). On the other hand, we did not observe clear cAMP changes in awake mice without any stimulation (Supplementary Fig. 6). These results indicate that not all the NAergic activities recruit astrocytic responses, but MP activities, particularly with longer duration, are tightly associated with astrocytic $Ca^{2+}$ elevations.

**Prolonged vigilance induces astrocytic cAMP increases.** To place our observations into a behavioral context, we imaged astrocytic $Ca^{2+}$ and cAMP dynamics in the auditory cortex of unanesthetized mice. Previous studies have shown that startle causes large $Ca^{2+}$ response in cortical astrocytes[9]. Therefore, we first tested whether a startle response drives not only $Ca^{2+}$ but also cAMP increases. In this experiment, mice received unpredictable air puffs onto the right side of the face for 10 s while the auditory cortex was imaged (Fig. 5a). In stark contrast to the expected astrocytic $Ca^{2+}$ elevations ($F/F_0$ 162.2 ± 15.6%) (Fig. 5b, d, e), we observed that air puffs did not lead to significant cAMP increases ($F/F_0$ 104.0 ± 1.7%) (Fig. 5c, f, g). These responses were qualitatively similar to the short stimulation of NA axons demonstrated in Fig. 1 (e.g. 3- or 5-s PS), suggesting that transient activation does not result in a long-lasting elevation of cAMP. To substantiate this premise, we imaged NAergic axon $Ca^{2+}$ activity during the course of the facial air puff experiment, which showed clear MP signals occur during abrupt air puff (Fig. 5h, i). In addition, we found that MP $Ca^{2+}$ activity evoked by air puff was significantly longer than that of before startle (Fig. 5j) (before: 2.84 ± 0.23 s, air puff: 3.57 ± 0.26 s). Taking a recent study that has shown astrocytic beta-adrenoreceptor activation leads to fear memory consolidation[14] into consideration, our results suggest

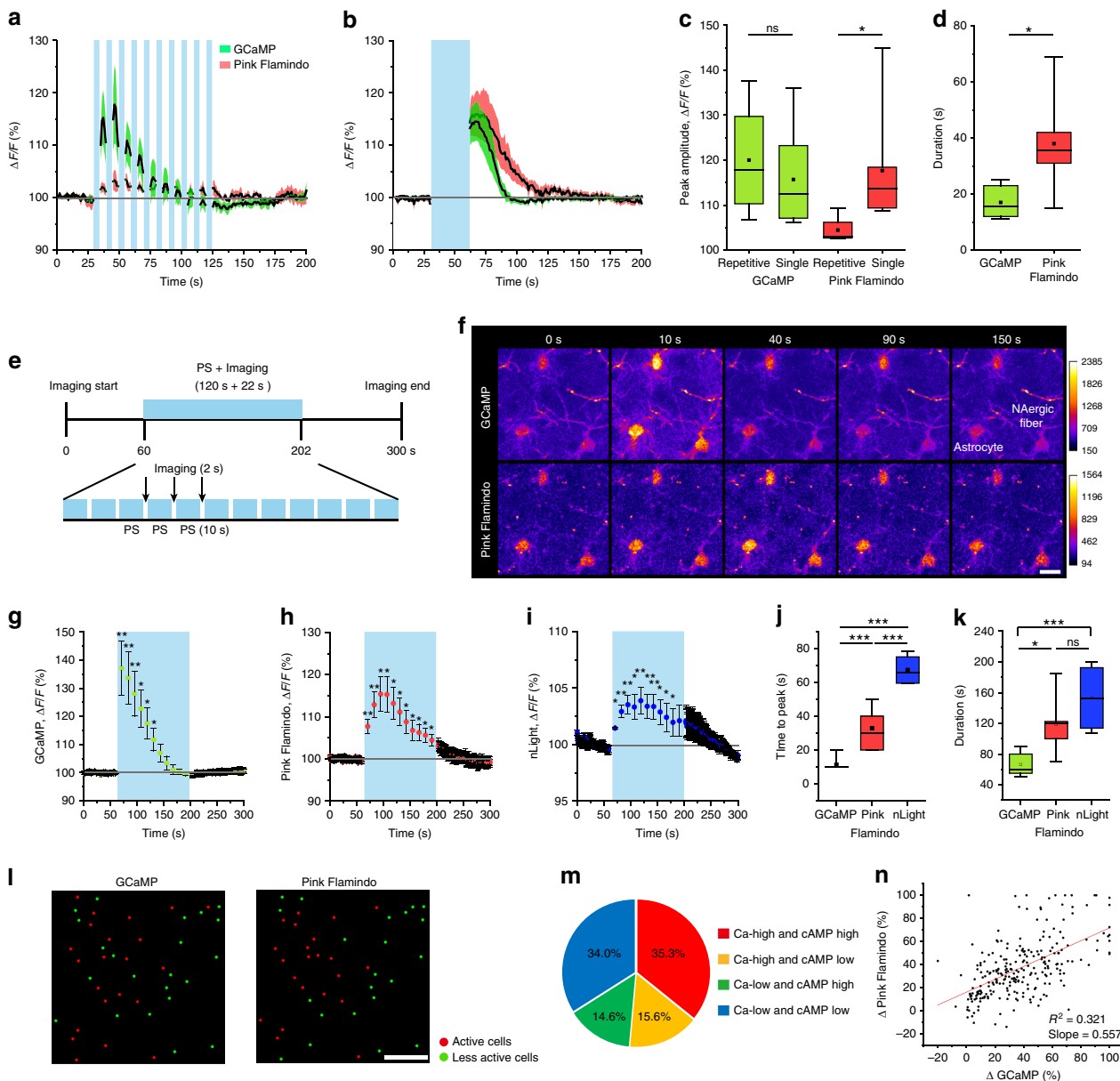

**Fig. 3 Distinct dynamics of astrocytic Ca$^{2+}$ and cAMP. a–d** Simultaneous imaging of GCaMP and Pink Flamindo with repetitive [3 s × 10, 7 s inter-stimulus interval] (**a**) and continuous [30 s] (**b**) PS. The peak amplitude of Pink Flamindo differs substantially between the two PS paradigms, whereas GCaMP shows similar peak amplitudes ($n = 4$–7 sessions, 4–6 mice) (**c**). Durations of astrocytic Ca$^{2+}$ and cAMP responses to 30-s PS are distinct (**d**) ($n = 6$ sessions, 4 mice). **e–k** 120-s PS of NAergic fibers. The long-time stimulation of NAergic fibers consists of 10-s PS and 2-s imaging that repeats 12 times (**e**). GCaMP signals show faster response and shorter duration (**f**, **g**), and Pink Flamindo signals exhibit slower increase and long-lasting duration (**f**, **h**). The dynamics of extracellular NA similarly long with that of astrocytic cAMP (**i**). Quantification of time to peak (**j**, $n = 4$–8 sessions, 4–6 mice) and duration (**k**, $n = 4$–8 sessions, 4–6 mice). Scale bar: 20 μm. **l–n** GCaMP and Pink Flamindo responses classified into active cells and less active cells (**l**). Red and green circles indicate active and less active cells, respectively. Combinatorial classification of GCaMP and Pink Flamindo responsiveness (**m**). Correlation of GCaMP signal changes and Pink Flamindo signal changes (**n**, $n = 249$ cells, 7 mice, Pearson correlation; $R^2 = 0.32$, slope = 0.557). Scale bar: 100 μm. Analysis was performed from average of individual sessions in **a–k**. **l–n** were analyzed from individual cell responses. Box plots: box range, 25–50–75% quatile; square, mean; whiskers, max–min. Student's $t$-test with Welch's correction: **c**, **d**; one-way ANOVA with Turkey's test: **c**; paired $t$-test: **j**, **k**; *$p < 0.05$, **$p < 0.01$, ***$p < 0.001$.

that a more prolonged behavioral paradigm is required for cAMP elevation in astrocytes, thus we next examined astrocytic activities during fear memory acquisition which includes prolonged periods of high vigilance.

We employed a cued fear conditioning paradigm in head-fixed mice to permit imaging of the cortex throughout the course of experiment (Fig. 6a). For fear conditioning, a foot shock (FS, 0.7 mA, 1 s) was delivered to the mouse during the last one second of the sound cue (Sound), whereas only Sound was presented during recall on the next day. Post-FS was further divided into post-FS1 and post-FS2 for later analysis (Fig. 6b). Electromyograph (EMG) from neck muscles showed a progressive decrease of muscle activity across repeated episodes of conditioning. The establishment of the fear memory was manifested by the increased immobility contingent to the presentation of Sound (Fig. 6c). We quantified immobility by

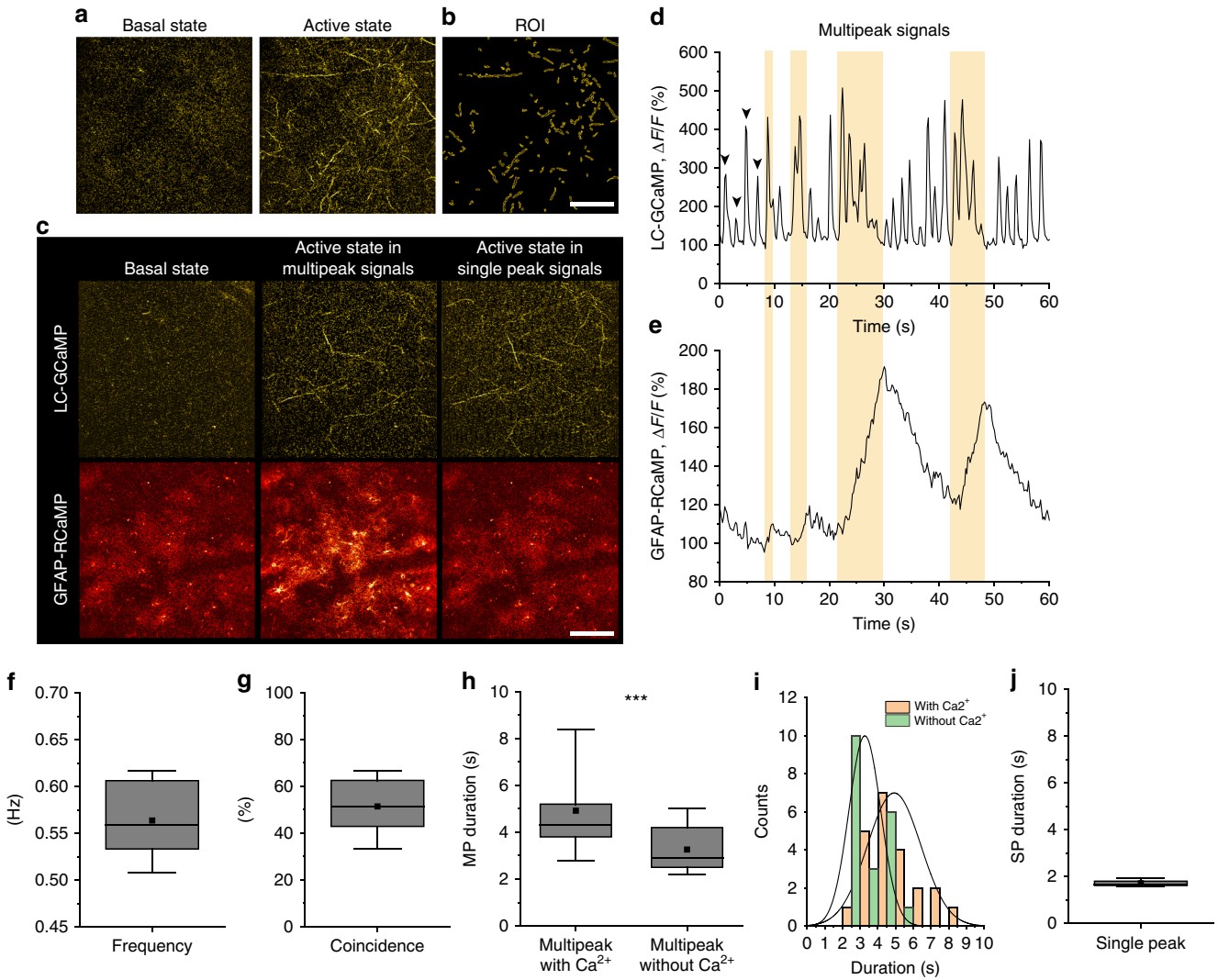

**Fig. 4 Astrocytic Ca²⁺ signals are accompanied by NAergic multipeak signals in awake spontaneous states. a, b** Representative images of a basal state and an active state of NAergic fibers' Ca²⁺ levels visualized by GCaMP6.f (**a**) and ROIs (**b**). Scale bar: 100 μm. **c** Representative pictures of LC-GCaMP and GFAP-RCaMP. Note that the GFAP-RCaMP images were acquired with a 5-s delay from the respective LC-GCaMP images. Only active states with MP signals accompany astrocytic Ca²⁺ elevation. Scale bar: 100 μm. **d, e** One minute simultaneous recording of LC-GCaMP and GFAP-RCaMP. NAergic signals are composed of SP signals (arrowhead) and MP signals (yellow period) (**d**). Astrocytic Ca²⁺ is correlated with NAergic MP signals (**e**). **f** Frequency of combined single and MP events. Individual MP signals were counted as single events ($n = 1276$ s recording, five mice for all analyses). **g** Coincidence of astrocytic Ca²⁺ with NAergic MP signals. **h, i** Durations of divided multipeak signals testing whether they accompany astrocytic Ca²⁺ are plotted in box plot (**h**) and histogram (**i**). **j** Durations of SP signals. Analysis was performed from individual SP or MP signals. Box plots: box range, 25–50–75% quatile; square, mean; whiskers, max–min. Student's t-test with Welch's correction: **h**; ***$p < 0.001$.

taking the ratio of EMG amplitude before vs. during the Sound phase. Accordingly, the immobility index significantly increased by the second conditioning trial and remained at this plateau throughout later conditioning (Fig. 6d) (1st: 0.95 ± 0.04, 2nd: 1.20 ± 0.10, 3rd: 1.64 ± 0.19, 4th: 1.23 ± 0.16, 5th: 1.29 ± 0.14, Recall-1st: 1.47 ± 0.17), indicating mice efficiently acquired the association of sound and FS. The next day presentation of the Sound alone evoked high immobility index, verifying memory formation (Fig. 6d).

Similar to the air puff, FS induced an elevation in Ca²⁺; however, the response attenuated with repeated shocks, with the amplitude decreasing to about 10% ΔF/F after second conditioning (Fig. 6e, g, i) (post-FS: 162.8 ± 12.8% (1st), 130.0 ± 7.0% (2nd), 111.8 ± 1.8% (3rd), 113.8 ± 3.5% (4th), 111.9 ± 3.4% (5th), 107.3 ± 1.6% (recall-1st)). In contrast to the transient facial air puff stimulation, cAMP elevation was discernible during fear conditioning (Fig. 6f, h). Astrocytic cAMP significantly elevated after the initial FS and then

slowly attenuated across trials (Fig. 6j) (post-FS: 110.9 ±2.9% (1st), 108.3 ± 2.3% (2nd), 105.4 ± 1.4% (3rd), 103.7 ± 0.6% (4th), 103.5 ± 0.6% (5th), 104.8 ± 1.0% (recall-1st)). The largest response was to the first conditioning (Fig. 6k, l) correlated with an observable change in the behavior of the mice (Fig. 6d). Similar to optogenetic NAergic activation, Ca²⁺ levels peaked faster than cAMP following the first FS (Fig. 6m, GCaMP: 7.0 ± 0.7 s, Pink Flamindo: 41.9 ± 12.8 s), indicating that the relatively slow and persistent effect of cAMP is similar under physiological conditions. Notably, Ca²⁺ and cAMP responses during fear conditioning were not observed in mice administered with a cocktail of NA antagonists (prazosin + propranolol) confirming the importance of NA input (Supplementary Fig. 7).

**High vigilance induced prolonged NAergic activity.** The results in Fig. 6 prompted us to examine the temporal organization of

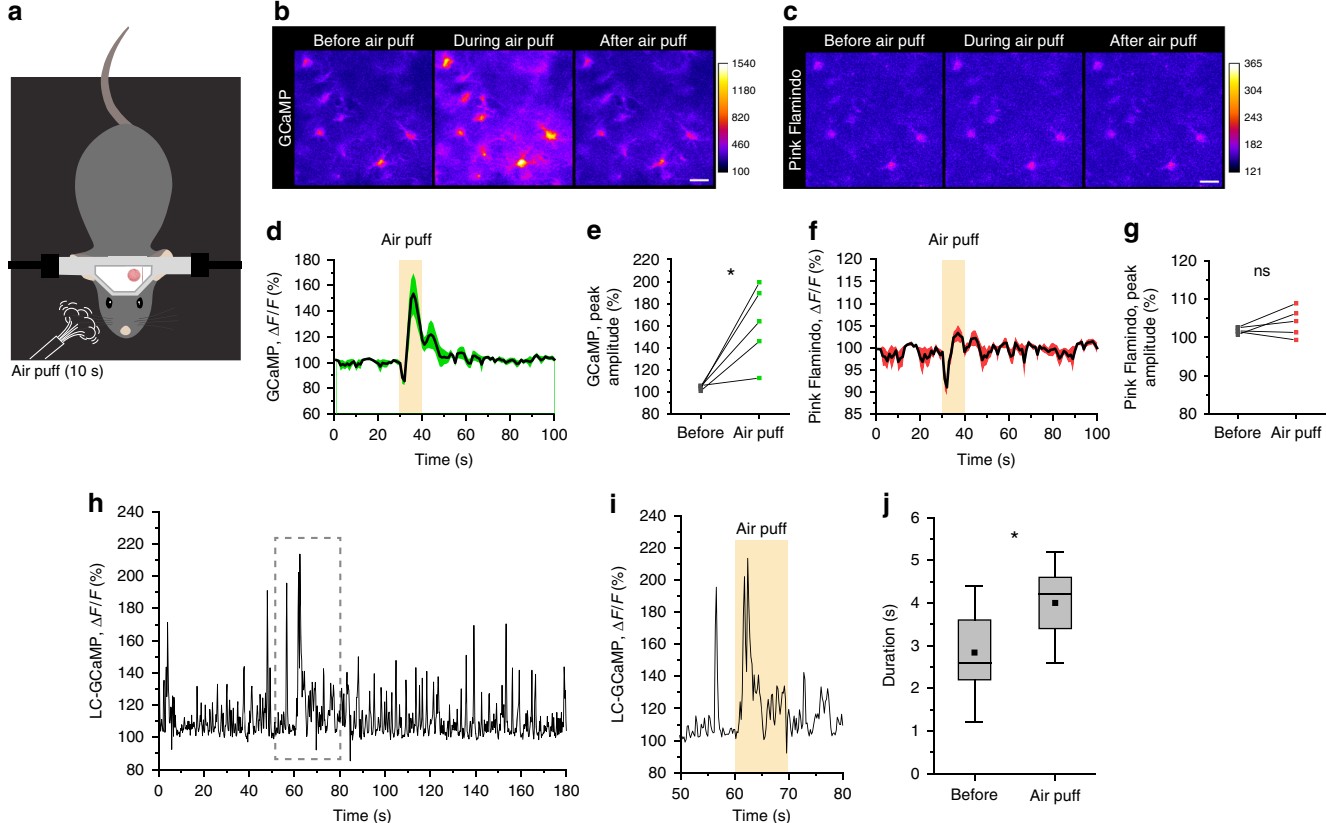

**Fig. 5 Startle by abrupt air-puff stimulation induces astrocytic Ca²⁺ elevations and LC/NA axon multipeak Ca²⁺ signals without a cAMP surge.**
**a** Illustration of air-puff stimulation under two-photon microscopy. **b, d, e** Astrocytes show clear Ca²⁺ elevations by facial air-puff stimulation (**b**). Air puffs presented for 10 s induce significant Ca²⁺ increases (**d, e**; $n = 5$ sessions, 3 mice). **c, f, g** Astrocytes do not show detectable cAMP signal changes by air puff ($n = 5$ sessions, 3 mice). **h, i** Ca²⁺ activity of LC/NA axons in the cortex was imaged for 3 min (**h**). The period during air puff was magnified (**i**), which shows a clear MP signal. **j** Duration of multipeak signals between before air puff (0–60 s) and during air puff (60–70 s) ($n = 4$ sessions, 4 mice). Scale bars 20 μm. Analysis was performed from average of individual sessions in **e** and **g** and from individual MP signals in **j**. Box plots: box range, 25–50–75% quatile; square, mean; whiskers, max–min. paired $t$-test: **e**; Student's $t$-test with Welch's correction: **j**; *$p < 0.05$.

NAergic activity during fear conditioning. LC/NA axon Ca²⁺ activity was imaged with the same paradigm as in Fig. 6. We found that MP signals were prevalent during the first ten seconds after the initial FS compared to the pre-FS period (Fig. 7a, c) (before: 2.82 ± 0.21 s, FS: 3.9 ± 0.54 s). However, this effect disappeared in later sessions (Fig. 7b, d) (before: 3.03 ± 0.28 s, FS: 3.07 ± 1.20 s), consistent with the reduced astrocytic Ca²⁺ response we observed. We further analyzed LC/NA axonal Ca²⁺ activity during the initial FS as illustrated in Fig. 6b. Whereas the mean of individual MP signal durations was similar in all phases (Fig. 7a, e), MP signals were observed more frequently during post-FS1 compared to the Control or Sound phase (Fig. 7a, f) (Control, Sound, post-FS, post-FS1, post-FS2: 3.6 ± 0.75, 3.6 ± 1.2, 7.4 ± 0.9, 9.0 ± 1.4, 5.8 ± 0.6 counts/min). This increase was not seen during post-FS2. Accordingly, the total time of MP signals increased significantly during post-FS1 (Fig. 7a, g) (Control, Sound, post-FS, post-FS1, post-FS2: 17.4 ± 4.1%, 16.8 ± 5.3%, 34.7 ± 3.4 s, 43.8 ± 5.5%, 25.7 ± 1.5%), but gradually attenuated in later sessions (Fig. 7h–l). Comparison of the first and fifth conditioning session revealed that a significant increase of NAergic MP signals occurred specifically in the post-FS1 period (Fig. 7m). These results are again consistent with the behavioral and cAMP responses that showed significant differences only following the first FS (Fig. 4d, l). In short, astrocytic Ca²⁺ elevations can be induced by a single NAergic MP activity while cAMP increase requires prolonged NAergic MP activity, indicating NAergic

activity can regulate two kinds of astrocytic second messengers depending upon the behavioral context. To test this hypothesis, we compared the frequency and total time of MP signals for startle and fear conditioning, confirming a prolonged effect spanning over a minute in fear conditioning (Supplementary Fig. 8).

Next, we again employed nLight to ask if the pattern of NAergic MP activity induced by fear conditioning increases extracellular NA. During behavior, the mean total time of MP signals were 17.5% and 43.8% in control and post-FS1, respectively. Thus, we mimicked this pattern optogenetically, employing two PS protocols, stimulating for 20% (3-s PS every 15 s × 5, sparse) or 42% (3-s PS every 7 s × 9, dense) of the total time across 1 min. The dense stimulation induced a mild, but significant increase of nLight signal, whereas the sparse stimulation did not lead to obvious changes of nLight signal (Fig. 7n, o, p) (Sparse: 101.2 ± 0.54%, Dense: 105.7 ± 1.5%) confirming that prolonged NAergic MP activity leads to an accumulation of extracellular NA and an elevation of astrocytic cAMP. Importantly, nLight imaging during fear conditioning induced reliable extracellular NA increases further lending support to the prevalence of NA signaling for the activation of astrocytes (Supplementary Fig. 9).

Finally, to investigate functional importance related to astrocytic cAMP signaling, we induced activated Gs signaling using a Gs-type DREADD (designer receptors exclusively activated by designer

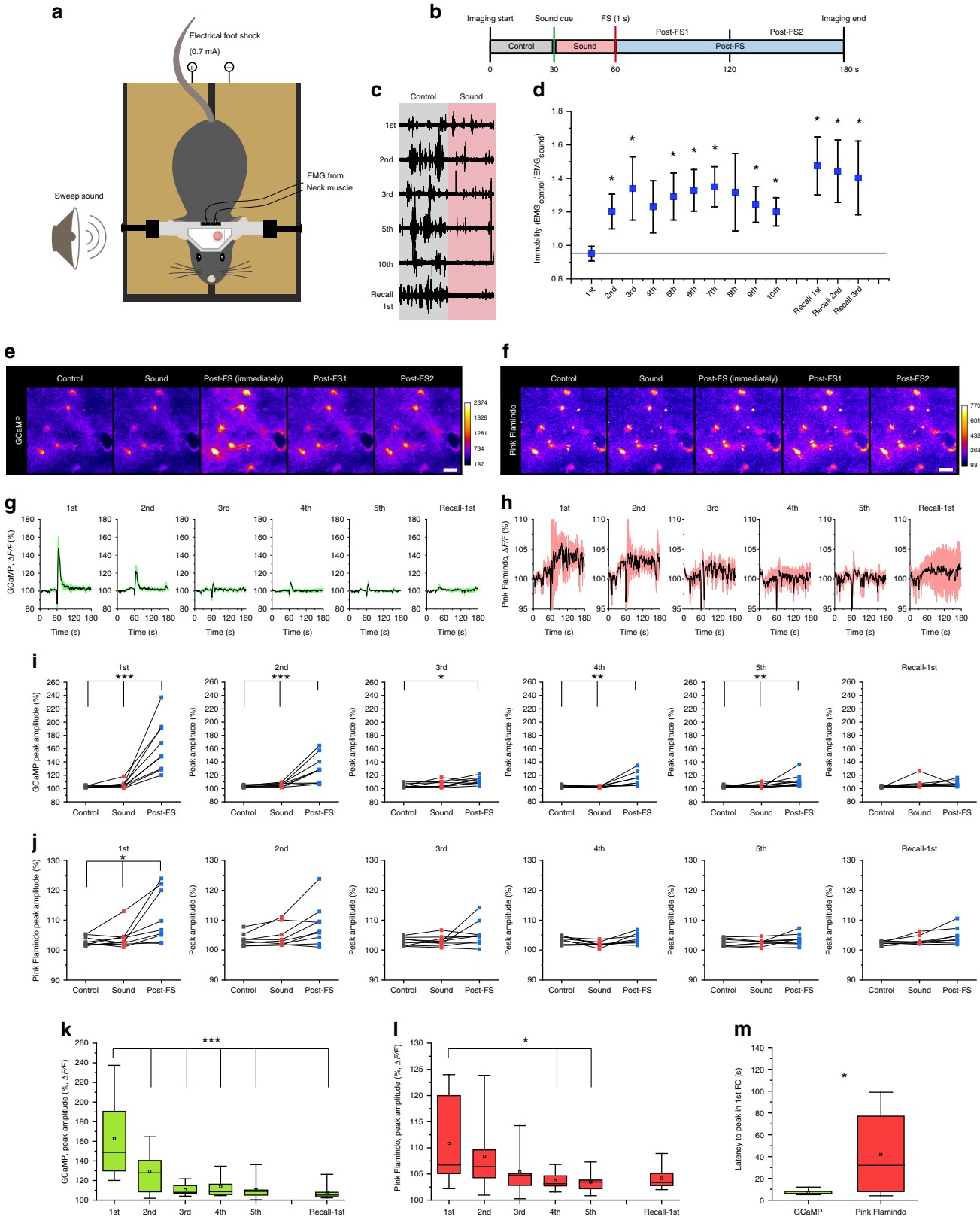

drugs) expressed in astrocytes via AAV8-GFAP-HA-rM3D-mCitrine[34] (Fig. 8a). A clear sustained cAMP elevation in astrocytes was observed by Pink Flamindo after CNO (Clozapine-N-Oxide) injection, lasting more than 2 h (Fig. 8b, c) (0 min: 99.2 ± 0.6%, 60 min: 180.6 ± 21.4%, 160 min: 174.9 ± 16.6%). Considering that one of the downstream functions of cAMP signaling in astrocytes is glycogenolysis, we visualized glycogen after CNO injection by immunohistochemistry. We confirmed that mice with rM3D-mCitrine expression in cortical astrocytes had distinctively lower levels of glycogen compared to control animals (Fig. 8d–f). These

**Fig. 6 High vigilance in head-fixed fear conditioning induces astrocytic Ca²⁺ and cAMP elevations. a** Illustration of head-fixed fear conditioning apparatus. **b** Protocol of head-fixed fear conditioning. **c** Representative EMG recording from neck muscles before and during sound cue. **d** Immobility index calculated by the ratio of EMG magnitudes before and during sound is plotted for each conditioning and recall session ($n = 8$ mice). **e, f.** Representative images of GCaMP (**e**) and Pink Flamindo (**f**) in astrocytes during the 1st conditioning. Images are averaged images of last 5 s from each phases except for post-FS (immediately). Post-FS (immediately) was averaged images of 3–7 s after PS (highest Ca²⁺). Scale bars: 20 μm. **g, h** Traces of mean ± SEM of GCaMP (**g**) and Pink Flamindo (**h**) in astrocytes for 1st–5th conditionings and 1st recall. **i, j** Peak amplitude of astrocyte GCaMP (**i**) ($n = 9$ mice for all analyses) and Pink Flamindo (**j**) in control, sound, and post-FS phase in 1–5th conditionings and 1st recall. **k, l** Comparison of peak amplitude in the post-FS phase of GCaMP (**k**) among 1–5th conditionings and 1st recall. **m** Latency to peak in the 1st conditioning shows a distinct difference between astrocytic Ca²⁺ and cAMP dynamics. Analysis was performed from the average of individual sessions. Box plots: box range, 25–50–75% quatile; square, mean; whiskers, max–min. Paired $t$-test (vs. 1st): **d**; one-way ANOVA with Turkey's test: **i–l**; Student's $t$-test with Welch's correction: **m**; *$p < 0.05$, **$p < 0.01$, ***$p < 0.001$.

results suggest a salient function of astrocytic cAMP is a boosting of energy metabolism mediated by glycogenolysis.

## Discussion

The present study is the first to visualize in vivo astrocytic cAMP dynamics in behaving animals. Earlier in vitro studies with cultured preparations have reported differences between Ca²⁺ and cAMP dynamics, finding that NA induces rapid Ca²⁺ and an order of magnitude slower cAMP elevation[35]. Here we employed optogenetic tools to assess Ca²⁺ and cAMP dynamics in the intact cortical circuit of live mice. Our data confirm the distinct temporal scales of the two second messengers in vivo in cortical astrocytes; however, we note that we did not observe consecutive repetition of Ca²⁺ surges which have been reported using in vitro preparations. The lack of oscillatory Ca²⁺ activity in astrocytes could be due to the transient presence of axon-released NA in the extracellular space, reflected in our optical measurements with nLight. We find that the duration of NA fibers stimulation is a key for the differential drive of astrocytic Ca²⁺ and cAMP activations; a short PS is sufficient for eliciting astrocytic Ca²⁺ elevation, whereas a longer PS is required for a cAMP increase. Our previous study demonstrated that Pink Flamindo can detect low concentrations of cAMP (~200 nM)[27] and we show here that a decrease of cAMP from the baseline is detectable after Gi pathway activation by hM4D DREADD (Supplementary Fig. 10). Considering that basal cAMP levels of mouse N1E-115 neuroblastoma cells are ~0.4 ± 0.3 μM and can increase up to 9 μM[36], the demonstration of both increase and decrease of Pink Flamindo signal warrants that our in vivo measurements of cAMP by this probe reflect physiological changes.

It has been reported that α2-ARs have the highest affinity for NA, followed by α1-ARs, and then β-ARs, which are coupled with the Gi, Gq, and Gs signaling pathway, respectively[19,37]. Such differential affinities can conceivably determine distinct thresholds for respective second messenger activation. In other words, a modest NA release is sufficient for α1-AR activation, whereas relatively high extracellular NA levels are needed for the increase of cAMP by β-AR activation. Extracellular NA accumulation was achieved by longer PS or prolonged PS of NAergic axons in our experiments, again supported by measurement of NA levels by nLight.

While 120-s PS of NAergic fibers led to extracellular accumulation of NA throughout the entire period, astrocytic Ca²⁺ elevations were not observable beyond 80 s. By contrast, steady increases of cAMP were observed in astrocytes during the PS. These distinct second messenger dynamics are presumably explained by the manner in which these respective second messengers are recruited. The α1-AR coupled Gq pathway induces a Ca²⁺ surge released from internal Ca²⁺ stores (e.g., endoplasmic reticulum). The self-amplifying mechanism of IP₃- and Ca²⁺-induced Ca²⁺ release presumably exhausts the Ca²⁺ in the internal store following continuous α1-AR activation. On the

other hand, Gs signaling produces cAMP from the abundantly available ATP in the cytosol via adenylyl cyclase without a regenerative mechanism; hence, it is continuously produced in the presence of sufficient extracellular NA to activate β-ARs (Supplementary Fig. 11).

Most analysis in the current study were performed in astrocytic cell bodies; however, a number of reports indicated astrocytic processes also exhibit Ca²⁺ activities[38,39]. Our analysis in Supplementary Fig. 12 showed that both Ca²⁺ and cAMP elevations were elicited also in astrocytic processes by activation of LC/NA axons with similar time courses to cell bodies but with smaller amplitudes. These results suggest that NA-induced GPCR signaling is induced at astrocytic processes. Notably, spontaneous transient microdomain Ca²⁺ activities recorded in wakeful conditions did not accompany detectable cAMP signals, whereas less frequent long-lasting Ca²⁺ activities accompanied mild cAMP elevation. Together with the positive correlation with Ca²⁺ duration, cAMP signals conceivably reflect the magnitude and spatial extent of extracellular NA level. Due to the close proximity of astrocytic processes to the synapse, astrocytic microdomain signaling has been implied in the modulation of synaptic signaling and plasticity. The wider and slower nature of cAMP signaling at the astrocytic process suggests that a spatiotemporally distinct mechanism of astrocyte–synapse interaction, possibly involving energy metabolism, takes place with cAMP elevation.

We examined the relationship between astrocyte second messenger dynamics and NAergic activity in awake mice. Previous work has found that startle or motor initiation evokes astrocytic Ca²⁺ elevation[8,9]. We could reproduce this startle-triggered astrocytic Ca²⁺ response by giving facial air puffs to the mouse at unpredictable timings; however, astrocytic cAMP responses were rarely observed. On the other hand, head-fixed fear conditioning induced significant astrocytic cAMP increases after FS, suggesting that the cAMP elevation occurs during prolonged, but not transient, vigilance states. This observation is consistent with previous studies reporting: (1) NAergic neurons are related to stress or aversive stimulation; particularly, FS induces phasic firing patterns in NAergic neurons[40]; (2) β-ARs play an important role in synaptic plasticity and memory[41,42]; and (3) blockade of astrocytic β-ARs impairs the memory formation of inhibitory avoidance[14]. Together these data suggest NAergic activity in a situation of high vigilance leads to astrocytic cAMP elevations via β-ARs and promotes memory formation. Moreover, in low vigilance states, astrocytic second messenger activity is predominantly Ca²⁺-oriented (Supplementary Fig. 13).

While we observed tonic Ca²⁺ events in NAergic axons, we found that astrocytic second messenger recruitment preferentially occurs during MP NAergic Ca²⁺ events. LC/NA neurons are known to elicit 1–3 Hz regular tonic firing in quiescent states and show bursting phasic firing with >10 Hz during arousal states[19]. Recent studies have reported that LC neurons exhibit

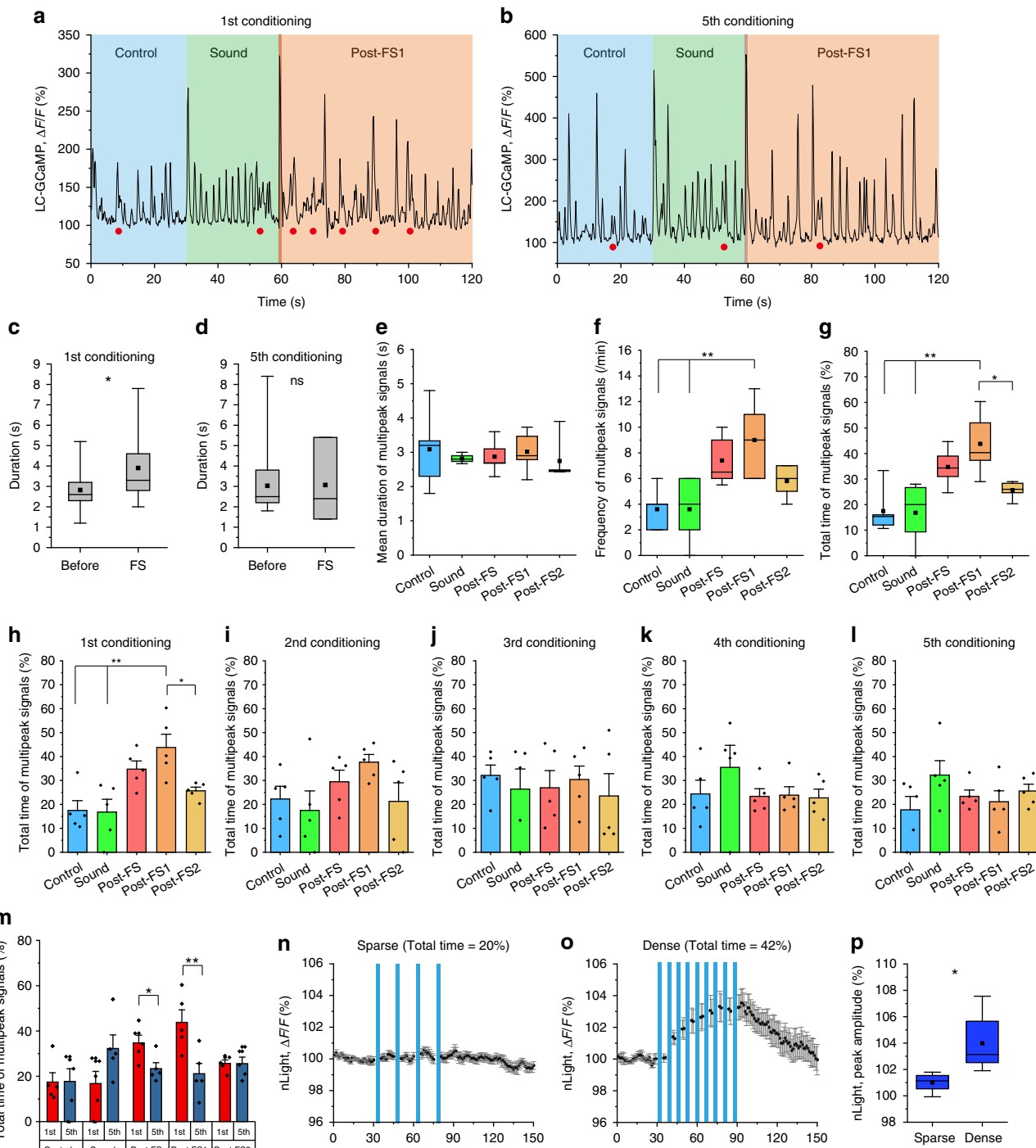

**Fig. 7 Head-fixed fear conditioning increases NAergic multipeak signals in 1st conditioning. a, b** Sample traces of NAergic fibers' Ca$^{2+}$ activity in 1st conditioning (**a**) and 5th conditioning (**b**). Filled red circles indicate MP signals. **c, d** Duration of multipeak signals were analyzed between before FS (0–60 s) and immediately after FS (60–70 s). Significant increase of duration was observed in the 1st conditioning, but disappeared in the 5th conditioning ($n = 5$ sessions, 5 mice). **e–g** Quantifications of NAergic multipeak signals: mean duration (**e**) ($n = 5$ mice for all analyses), frequency (**f**), and total time (**g**). **h–l** Total times of MP signals in 1–5th conditionings are shown. Post-FS1 has significantly longer total time only during 1st conditioning. 1st (**h**), 2nd (**i**), 3rd (**j**), 4th (**k**), and 5th conditioning (**l**). **m** Comparison of each conditioning session between 1st and 5th conditioning. Increased total time of post-FS, particularly post-FS1, in 1st conditioning disappear in 5th conditioning. **n–p** mimicking NAergic fiber activation during the head-fixed fear conditioning show that dense PS (**o, p** total time = 42%, corresponding to post-FS1 phase) significantly increases the extracellular NA concentration than sparse PS (**n, p** total time = 20%, correspond to control phase; $n = 6$ sessions, 4 mice). Analysis was performed from individual MP signals in **c** and **d** and average of individual sessions for other graphs. Bar graphs: mean + SEM. Box plots: box range, 25–50–75% quatile; square, mean; whiskers, max–min. Student's $t$-test with Welch's correction: **c, d, p**; one-way ANOVA with Turkey's test: **f–h**; paired $t$-test: **m**; *$p < 0.05$, **$p < 0.01$.

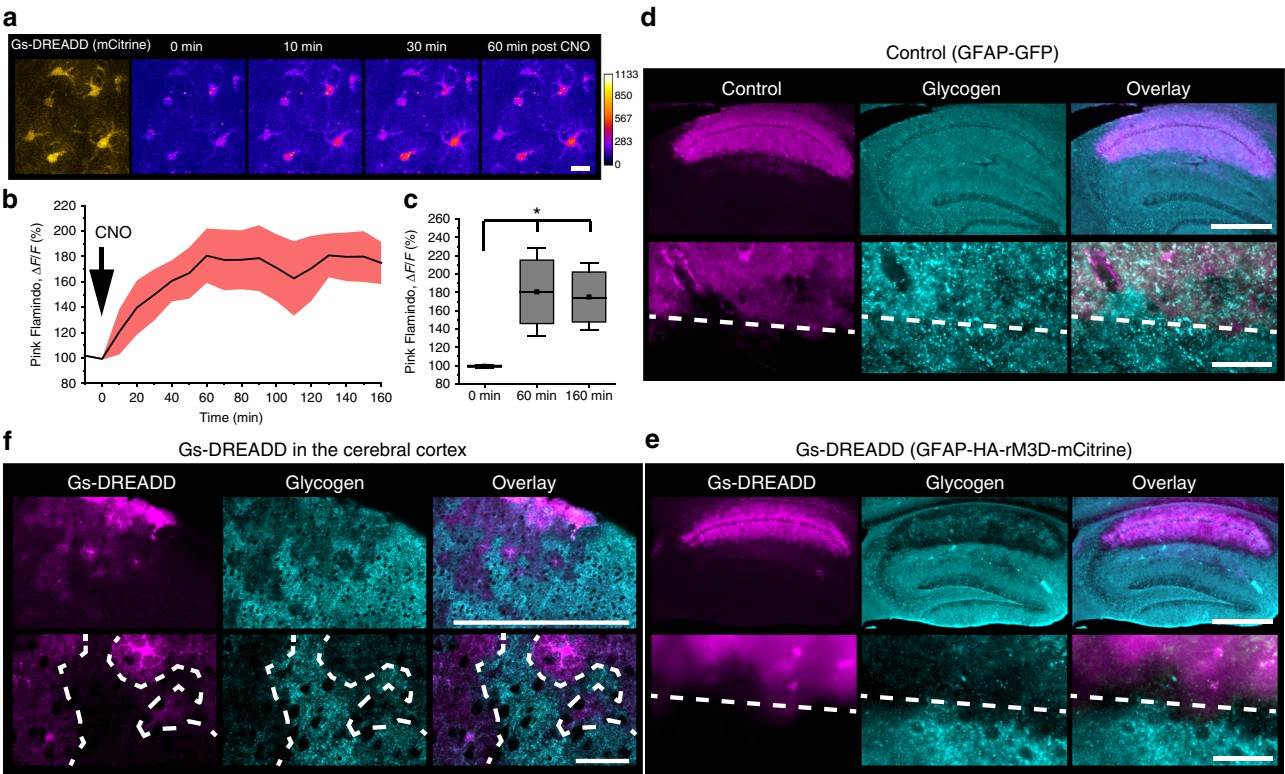

**Fig. 8 Chemogenetic elevation of astrocytic cAMP induces glycogen breakdown. a** Representative images of Gs-DREADD with mCitrine (yellow) and Pink Flamindo (pseudocolor) with CNO injection (1 mg/kg, i.p.). Pink Flamindo and Gs-DREADD (rM3D) were expressed in cortical astrocytes by co-injection of the respective AAVs. Scale bars: 20 μm. **b**, **c** Time course of Pink Flamindo signal change (**b**). Comparison of Pink Flamindo signal shows that the effect of Gs-DREADD lasts at least 160 min (**c**) ($n = 4$ mice, one-way ANOVA, $p = 0.0087$: Tukey's test,\*$p < 0.05$). **d–f** Glycogen immunohistochemistry on control (GFAP-GFP) (**d**) and Gs-DREADD (GFAP-HA-rM3D-mCitrine) in the hippocampus (**e**) and in the cortex (**f**) fixed 120 min after CNO injection. Lower images are high magnification of the upper images. White dash lines indicate the border of virus infection. Scale bars: 500 μm (upper) and 50 μm (lower). Analysis was performed from average of individual sessions. Box plots: box range, 25–50–75% quatile; square, mean; whiskers, max–min. One-way ANOVA with Turkey's test: **c**; \*$p < 0.05$.

target-specific projections[43,44] and show differential discharge rates depending on target area[45]. For instance, the orbitofrontal and medial prefrontal cortices have the highest NAergic activity with approximately 1.5 Hz and the primary motor cortex has the lowest activity with 0.5 Hz. Our axonal Ca$^{2+}$ imaging yields tonic SP events in the range of the previously reported LC neurons. Thus, the basal SP axonal Ca$^{2+}$ events likely represent regular tonic firing. Furthermore, we suppose that MP signals reflect burst firing, since spikes in a burst result in step-wise GCaMP.6f fluorescence increases[29].

In summary, we propose that a startle response triggers phasic firing in NAergic axons which evokes astrocytic Ca$^{2+}$ elevation, while a prolonged vigilance causes continual phasic NA activities which gradually induce astrocytic cAMP elevations. Our extracellular NA measurement with nLight reinforces the notion that NAergic activity patterns are a major determinant of the extracellular NA concentration and the main factor for astrocytic second messenger recruitment.

A number of in vivo studies have implicated supportive roles of astrocytic Ca$^{2+}$ elevation in synaptic plasticity and memory following neuromodulator-driven[5,6,46–48] or astrocyte-targeted synthetic GPCR activation[12,49]. On the contrary, astrocytic IP$_3$ receptor type 2 (IP$_3$R2) knockout mice, which lack Ca$^{2+}$ increase in astrocytes, do not exhibit any behavioral differences, including in learning tasks[11]. Thus, the essential function of astrocytic Ca$^{2+}$ remains controversial.

Behavioral influences of astrocytic cAMP are relatively less understood. Alberini's group[14] used pharmacological blockade of

β2-ARs or genetic knockdown of β2-ARs in astrocytes by shRNA in the hippocampus and demonstrated the impairment of memory consolidation in an inhibitory avoidance task. Such a reduction of long-term memory was rescued by administration of lactate, suggesting that astrocytic β2-AR activation results in glycogenolysis that promotes lactate transfer to neurons. Though our measurements performed in the auditory cortex, our results are consistent with this study in that cAMP, the downstream of target of β2-AR Gs signaling, is elevated during vigilance and learning. However, our pharmacological experiments indicated that β1-AR is the major functional subtype for the elevation of astrocytic cAMP. One possible explanation is the differences of animal species and brain region: our study was in the mouse cortex, whereas Alberini's group investigated the rat hippocampus. A previous report had shown that expression ratio of β1- or β2-AR depends on species[31]. For instance, β1-AR is the major adrenoreceptor subtype in mouse cortical astrocytes, whereas the ratio is equal in human astrocyte culture.

It is noted that Mucke's group[16] has shown that chemogenetic activation of astrocytic Gs-GPCR RS1 reduces Morris water maze performance. According to this study, constitutive RS1 activity was sufficient to introduce a mild degree of compromise in memory consolidation, and ligand activation of RS1 lead to further memory deficit. In light of our finding that astrocytic Gs activation by DREADD dramatically reduced glycogen pool, it is possible that the constitutive and ligand-induced RS activation resulted in a decline and depletion of glycogen in the astrocytes.

Glycogen in astrocytes is an emerging factor in memory consolidation. While astrocytes are the cell type that stores appreciable amounts of glycogen, their glycogen storage is heterogeneous in that there are glycogen-rich astrocytes and glycogen-poor astrocytes distinguished by the presence of glycogen large-particles[50,51] In the current study, we showed the composition of adrenoreceptor subtypes is likely heterogeneous in individual astrocytes. Since astrocytic cAMP promotes glycogen metabolism, this may underlie the heterogeneous glycogen distribution.

The diversity of astrocytic GPCR systems suggests that astrocyte–neuron interactions are mediated by the concordant action of multiple GPCR signaling pathways. Our results reveal two temporally distinct second messenger dynamics that occur in a context-dependent manner and elucidate how a neuromodulator can tune the astrocyte–neuron interactions: Rapid and transient astrocytic $Ca^{2+}$ could play important roles in sub-minute scales which might involve modulation of synaptic transmission, and slow and long-lasting astrocytic cAMP could be involved more in brain functions of longer time scales, such as consolidation of memory mediated by energy metabolism.

## Methods

**Surgery**. Adult mice (postnatal 2–4 months) were anesthetized with ketamine and xylazine (70 and 10 mg/kg, respectively, i.p.) for introduction and kept under stable anesthesia with isoflurane (0.5–1.0%) until the end of surgery. Surgery was performed using a stereotaxic apparatus. After a small cranial hole was made above the cerebellum at the stereotaxic coordinate of AP −5.5 mm, ML +0.9 mm by a dental drill. Intracranial microinjection of AAV (AAV-DJ/8-EF1a-DIO-ChR2-EYFP, AAV-DJ/8-EF1a-DIO-GCaMP6.f, or AAV-DJ/8-EF1a-DIO-EYFP ($1–3 \times 10^{12}$ vg/mL) was conducted using a glass micropipette connected to a Femtojet microinjector (Eppendorf) at three depths (2.5, 3.0, and 3.5 mm from the surface, 300 nL at each location). Microinjection of AAV to the cerebral cortex was performed after attachment of a stainless headplate to the skull with dental cement. Microinjection (300 nL) of AAV9-GFAP-GCaMP7.09 ($3.0 \times 10^{12}$ vg/mL), AAV9-GFAP-Pink Flamindo ($6.6 \times 10^{12}$ vg/mL), AAV9-GFAP-RCaMP1.07 ($3.0 \times 10^{12}$ vg/mL), or AAV9-hSynI-nLight ($1.0 \times 10^{13}$ vg/mL) was made in the auditory cortex (AP −2.0 to −2.5 mm, ML +3.0 to 4.0 mm, DV −0.3 mm) or the parietal cortex (AP −1.5 to −3.0 mm, ML 1.5 to 3.5 mm, DV −0.3 mm). The virus-injected area was covered by a sterilized round cover glass (3 or 4 mm in diameter) to serve as a cranial window for two-photon imaging. After surgery, mice were kept on a heat pad for recovery (37 °C, 2 days). Allen Mouse Brain Atlas was referenced for illustration of virus injection in Fig. 1a (https://mouse.brain-map.org/static/atlas)

For head-fixed fear conditioning experiments, additional surgery for electromyography (EMG) electrode implantation was performed. One week after the preparation of cranial window, tungsten wires (50 μm in diameter) were inserted in neck muscles and fixed with dental cement. After surgery, mice were kept on a heat pad for recovery (37 °C, 2 days).

The AAV-expressed fluorescent indicators were observable after 2 weeks. For ChR2-EYFP, we waited at least 3 weeks, so that sufficient amounts of ChR2 protein are expressed in the LC-originated NAergic fibers for cortical PS. For head-fixed fear conditioning experiments, we performed 5 days of handling and habituation (20–40 min per day for individual mice) for wild-type mice and 10 days for NET-cre mice before imaging.

**Imaging and head-fixed fear conditioning**. Two-photon imaging was performed using a B-scope (Thorlabs) with a Chameleon Vision 2 laser (Coherent) or a Bergamo (Thorlabs) with a Chameleon Ultra 2 laser (Coherent). For imaging of GCaMP or nLight alone, an excitation wavelength of 920 nm was used. For single or dual imaging involving Pink Flamindo or RCaMP, the excitation wavelength wad adjusted to 1020–1060 nm. Emission was separated by a 562 nm dichroic mirror (Semrock) with a 525/50 nm band pass filter (Semrock) or a 607/70 nm band pass filter (Semrock). Images were acquired using the ThorImage$^R$LS software. In all imaging, image size was $400 \times 400$ μm ($512 \times 512$ pixels), and XLPlan N ($\times 25$ NA = 1.05; Olympus) or Apo LWD ($\times 25$ NA = 1.10; Nikon) was used as the objective lens.

GCaMP and Pink Flamindo imaging with NAergic fiber PS was performed in isoflurane-anesthetized (1.0–1.5%) mice. Mice were rigidly fixed in a headplate holding device with an angle adjuster (MAG-2; Narishige) and placed under a two-photon microscope. PS was composed of 10 Hz pulses of 50 ms width and controlled by a pulse generator (Master-8, A.M.P.I). Stimulation of ChR2-expressing NAergic axons was performed by an LED light source directed to the cranial window area through the objective lens of the two-photon microscopy. To prevent photo damage of photomultiplier tubes (PMTs) during PS, the optical path to the PMTs was blocked by a shutter.

Adrenoreceptor subtypes were blocked by the following reagents: prazosin (10 mg/kg; Sigma-Aldrich) for α1-AR, propranolol (10 mg/kg; Sigma-Aldrich) for non-selective β-ARs, betaxolol (10 mg/kg; Tokyo Chemical Industry Co., Ltd.) for β1-AR, and ICI 118,551 (20 mg/kg, Tocris) for β2-AR.

For sound-cued head-fixed fear conditioning experiments, mice were briefly anesthetized by isoflurane to fix the head to the holding device. Mice were kept with the holding device until anesthesia wears off (30–60 min) before imaging. Images of simultaneous astrocytic $Ca^{2+}$ and cAMP were acquired at 1 Hz. Images for 120-s PS experiment and LC-GCaMP imaging experiments were acquired at 2 and 5 Hz respectively. Sound cue was composed of sweep-up sound (0–40 kHz, 2 s) programed by MATLAB and LabVIEW and was continuously repeated for 30 s for each conditioning and recall. During the last second of sound cue, a single FS (0.7 mA, 1 s) was given through a metal plate connected to a stimulus isolator (ISO-Flex, A.M.P.I) and a pulse generator (Master-8). EMG from neck muscles was amplified by an ELC-03XS amplifier (NPI) and digitized at 1 kHz by the Thorsync software. The recall experiment procedure was essentially a conditioning procedure without the FS.

In a subset of nLight imaging experiments, a selective NA reuptake inhibitor desipramine (10 mg/kg; Sigma-Aldrich) was injected intraperitoneally 30 min before imaging and compared with images taken before the drug application. Images were acquired at 2 Hz.

For Pink Flamindo imaging with Gs-DREADD activation in astrocytes, AAV9-GFAP-Pink Flamindo and rAAV8-GFAP-Ha-rM3D-IRES-mCitrine were co-injected in the parietal cortex. After 2 weeks, we first imaged mCitrine with 960 nm wavelength beam to determine the expression of Gs-DREADD. Thereafter, Pink Flamindo imaging was performed with 1060 nm wavelength laser for cAMP dynamics. Thirty-micrometer-depth imaging with 1 μm steps were done at every 10 min. A total of 3 h imaging was performed including the control period of pre-CNO injection (10 min).

All experiments with awake mice were performed in the auditory cortex, whereas experiments involving optogenetic activation of LC-NA axons were performed in the parietal cortex.

**Histology**. Virus injection of AAV-DJ/8-EF1a-DIO-EYFP in the LC was performed as described above. Two weeks after virus injection, mice were sacrificed by transcardiac perfusion. Briefly, 25 mL saline (0.9% NaCl) was perfused for 5 min, followed by 50 mL of fixative (4% paraformaldehyde in 0.1M phosphate buffer, PFA-PB) for 10 min. Perfusion-fixed brains were post-fixed overnight in PFA-PB. Sagittal brain sections of 60 μm thickness were prepared in PB using a microslicer (Pro-7 Linear Slicer, DSK, Japan). After washing in phosphate-buffered saline (PBS), the sections were incubated in PBS containing 0.1% Triton X-100 and primary antibodies were applied overnight at 4 °C while gently shaking. The concentrations of primary antibodies were as follows: polyclonal anti-GFP[52] 1:1000 and monoclonal anti-TH (Merck Millipore, ms) 1:1000. The sections were then washed three times in PBS and incubated with fluorescent secondary antibodies (1:1000, in PBS containing 0.1% Triton X-100, Alexa Fluor 488 or 594; Life Technologies). Brain slices were mounted on slide glasses and coverslipped with VECTASHIELD (Vector laboratories). The sections were imaged by a Keyence all-in-one microscope (BZ-9000). Images were acquired with a z-step of 10 μm, using a ×10 objective lens (Nikon, Plan Apo NA = 0.45).

For glycogen staining, AAV9-GFAP-GFP or rAAV8-GFAP-Ha-rM3D-IRES-mCitrine-injected mice were sacrificed by focused microwave irradiation (5 kW for 1 s) 2 h after CNO (1 mg/kg, i.p.) injection. Briefly, after incubating the brain in 4% PFA overnight, sections containing the hippocampus was sliced with 60 μm thickness. Thereafter, sections were treated with anti-heat-denatured GFP antibody[53] and anti-glycogen antibody (IV58B6)[54] for 24 h, followed by 2 h incubation in secondary antibodies with Alexa Fluor 488 and Alexa Fluor597, respectively[50].

**Analysis**. Imaging analysis was performed by ImageJ and MATLAB. Image shift in xy-axis was adjusted by the Turboreg ImageJ plug-in program for all images. ROIs of GCaMP and Pink Flamindo signals in astrocytes were extracted from cell bodies manually on ImageJ, and the intensity data were exported for further analysis in MATLAB. Peak amplitude, duration, and time to peak were calculated from ΔF/F, where F is the mean fluorescence intensity within a given ROI during the control period. Duration was defined as the time to reach at 80% signal reduction from peak. Time to peak was defined as the time reach to peak after PS. For 10 s × 12 PS experiments, images were obtained during the 2-s interval between individual PSs with 2 Hz acquisition rate, and the frames in the 2-s interval were averaged. The threshold for active vs. less active cells regarding $Ca^{2+}$ and cAMP responses after 30-s PS was set at the median; therefore, the top 50% was defined as active cells and bottom 50% was defined as less active cells. Cell positions were defined as the center of individual ROIs. The values used for correlation of ΔGCaMP and ΔPink Flamindo were normalized by maximum ΔF/F values as 100%.

In fear conditioning experiments, immobility was calculated as the ratio of EMG magnitude in during control and the sound phase. The initial five trials of conditioning and the first trial of recall were analyzed. Peak amplitude was extracted individually from control, sound, and post-FS phases. Latency to peak was defined as the time to reach to peak after FS. Images of GCaMP.6f in LC/NA

axons were acquired at 5 Hz, since pilot experiments showed that most of LC/NA axon GCaMP6.f activity is captured at 2 Hz sampling (Supplementary Fig. 5). Since the background pixel noise associated with fast image acquisition misrepresent the mean image, the median image was taken as a reference. Therefore, the apparent $\Delta F/F$ is amplified than that calculated with the raw image. The processed images were averaged, and then binarized with a threshold of 10% maximum fluorescence intensity. ROIs for NAergic axons were extracted from the above-threshold pixels with the following conditions; the contiguous pixel number is more than 30 and the circularity is less than 0.5. MP signals were automatically detected by a custom-written MATLAB program. Briefly, the trend was removed from the GCaMP $\Delta F/F$ signal. Thereafter, local minima of the baseline-adjusted $\Delta F/F$ signal are calculated and the standard deviation of local minima $\sigma_{LM}$ was computed. MP signals are detected as the periods that have multiple $\Delta F/F$ peaks whereby the $\Delta F/F$ local minima stay above baseline $+1.5 \times \sigma_{LM}$.

When RCaMP in cortical astrocytes was imaged with LC/NA axonal GCaMP for awake spontaneous activities, coincidence of RCaMP and GCaMP events were counted if an RCaMP events appear within 10 s from the beginning of a MP signal. For Ca$^{2+}$ event frequency counts, both MP signals and MP signals were counted as single events (i.e. individual signal peaks within an MP event was not counted). Duration of MP signals and MP signals were computed by MATLAB.

ROIs for nLight were identified from binarized image. Briefly, raw image was binarized with a threshold of 50% maximum fluorescence intensity, followed by removal of LC fibers' occupancy labeled with ChR2-EYFP. ROIs were detected from such binarized image with following parameters; contiguous pixel number >50 and 0 < circularity < 1. $\Delta F/F$, Peak amplitude, duration, and time to peak were calculated in the same manner as GCaMP or Pink Flamindo.

Regarding Pink Flamindo imaging with Gs-DREADD, 10 frames from 30 μm image stacks were used. Corresponding depth for all images were selected by visual inspection. $\Delta F/F$ values were calculated from stacked images by the maximum intensity Z projection of 10 frames.

**Statistics**. All statistical analyses were done in origin. Comparisons between two groups were analyzed with Student's t-test with Welch's correction or paired t-test. For comparisons of data between before and after a manipulation, paired t-test was applied. For other comparisons of two sample means, Student's t-test with Welch's correction was used. Comparisons of three or more groups were performed using one-way ANOVA with post hoc Tukey's test. Detailed experimental values are shown in the main text. Most plots were drawn by origin except for pie chart (Excel) and some figures were made by MATLAB (Figs. 1l and 2k).

**Ethics declarations**. The procedures involving animal care, surgery, and sample preparation were approved by the Animal Experimental Committee of RIKEN Brain Science Institute and performed in accordance with the guidelines of the Animal Experimental Committee of RIKEN Brain Science Institute/Center for Brain Science.

**Reporting summary**. Further information on research design is available in the Nature Research Reporting Summary linked to this article.

## Data availability
The data that support the findings of this study are available from the corresponding author upon reasonable request.

## Code availability
The code used in this study is available from the corresponding author upon reasonable request.

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

## Acknowledgements

This work was supported by the RIKEN Brain Science Institute, KAKENHI grants 26117520, 16H01888, 18H05150 (to H. Hirase), 19H01012, 19H05233, 19H05646 (to T.J.M.H.), HFSP RGP0036/2014 (to H. Hirase), NIH BRAIN Initiative U01NS090604 and U01NS013522, DP2MH107056 (to L.T.). We thank Prof. Maiken Nedergaard for comments, Dr. Tetsuya Kitaguchi for providing Pink Flamindo construct, and members of the laboratory for comments on earlier versions of the manuscript. This research is partially supported by the program for Brain Mapping by Integrated Neurotechnologies for Disease Studies (Brain/MINDS) from Japan Agency for Medical Research and development, AMED under the Grant Number JP19dm0207057 (to H. Hirai).

## Author contributions

Y.O. and H. Hirase designed the study and contributed to analysis. Y.O. contributed to virus injection, surgery, and histology. Y.O. and X.W. contributed to two-photon imaging. T.P. and L.T. contributed to development of nLight. T.J.M. generated NAT-Cre mice. A.K., H. Hirai, and T.J.M. produced AAVs. K.O. set up two-photon microscopy with LED photostimulation. K.Y. performed animal care. Y.O. and H. Hirase wrote the manuscript with input from all the authors.

## Competing interests

The authors declare no competing interests.
