## [Peer Review File · Nature Communications]

Reviewers' Comments:

Reviewer #1:

Remarks to the Author:

Combining optogenetics, behavioral tools and two photon imaging technique the authors show that:

- Astrocytic Ca²⁺ and cAMP response from parietal cortex by NAergic axon photostimulation are regulated in a temporally distinct manner.
- Astrocytic cAMP dynamics aligns with extracellular NA in the cortex.
- Facial air-puffs induce astrocytic Ca²⁺ signaling in the cortex and foot shock paradigm induces astrocytic cAMP and Ca²⁺ response in the cortex. Both signaling exhibit different temporal curve. Based on these observations, the authors conclude that distinct astrocytic signaling pathways can integrate noradrenergic activity during vigilance states to mediate distinct functions supporting memory.

The experimental design and the results presented in the manuscript are interesting, simple and straightforward. The findings have novelty, confirming that astrocytes exhibit different functional properties. This study is carefully carried out. However, I have a several concerns that the authors need to address to fully support their conclusions. These are listed below:

1- A critical aspect of the manuscript is based on the fact that cAMP signaling is typically triggered by a rather non-physiological manipulation, namely, 10 s photostimulation. What endogenous activity is this photostimulation mimicking? Under what physiological circumstances do astrocytes show similar activity as reported here by artificial activation? Such induction protocol raises the key question: In what physiological conditions does this phenomenon happen? Although the author showed input/output function to determine what kind of activity is required to triggers, the authors must show that action potentials can trigger these phenomena and to prove that it is not a technical artifact of the experimental approaches used in this study.

2.- How does NA activate astrocytes? How does it work? What kinds of receptors are activated? In the abstract section the authors stated: "α1-adrenergic receptors triggers rapid astrocytic Ca²⁺" and "β-adrenergic receptors elevates cAMP levels". The authors should check these statements with experimental data as part of this study (for example, using pharmacological approaches).

3.- I am confused regarding the absence of dLight responses to low extracellular NA concentrations. In my view the paper would gain in clarity if they could show dLight doses-response curve, applying different know concentration of NA and DA. Fig 3B. What endogenous activity is this photostimulation mimicking?

4.- Along the same line, does calcium signaling come from the cell soma, from the cell processes, or are they an average? Include details on how the ROIs were generated (both in neurons and in astrocytes). It is a key question after Volterra ´s lab work: Bindocci et al., Science 2017. See points 1 and 3.

5.- Figure 4. What about cAMP in awake animals? What about astrocytic cAMP in basal states? Is there any correlation between NAergic multipeak Ca²⁺ signals and astrocytic cAMP signals? These results should be shown in the figure.

6.- Page 6. The authors indicate "The next day presentation of the sound alone evoked high immobility index, verifying memory formation". This is an important statement. However, I don't find

the experimental data.

Minor points

7.- At times, the manuscript is difficult to read. It is useful explain the data and removed the percentage numbers.

Reviewer #2:

Remarks to the Author:

In this paper, the authors investigated the response of astrocytes - in terms of both Ca²⁺ and cAMP signaling- to the synaptic release of NA induced in vivo by optogenetic stimulation of locus coeruleus (LC) NA neurons and during air puff-induced startle response. They also studied the two signals in fear conditioning with a foot shock protocol. In optogenetic experiments the authors showed that the cAMP signal response displays higher activation threshold, slower rise time and longer duration with respect to the calcium signal response. Using the sensor dLight the authors could confirm that the cAMP response is associated with accumulation of NA in the extracellular space.

The use of G-CaMP6f expressed in NAergic neurons revealed a correlation between their firing pattern and astrocytic calcium response, with longer duration multiphase (MP) signals in neurons co-occurring with Ca²⁺ surges.

Air puff-induced startle caused large calcium responses, but not cAMP increases, in cortical astrocytes. Accordingly, calcium MP signals evoked in NAergic neurons by startle were significantly longer than before startle.

The authors chose then a paradigm able to induce high vigilance states to check for cAMP responses during fear memory acquisition. Foot shock paradigm induced both calcium and cAMP responses, again with different dynamics. The largest response of the second messengers was observed upon the first conditioning and correlated with an observable change in the mouse behaviour. This response declined upon further conditioning sessions. Duration and frequency of MP signals in NAergic neurons exhibited similar dynamics, with total time of MP signals significantly increasing during the first conditioning and recovering in the successive sessions.

The authors then employed dLight to investigate the increases in extracellular NA occurring during MP activity induced by fear conditioning. To this aim, they mimicked the pattern of MP signals before and after FS through optogenetic stimulation, revealing that only dense stimulation resembling MP signals pattern in post-FS1 phase induces a mild but significant NA increase in the extracellular space, consistent with the cAMP response. Finally, the authors used Gs-type DREADD selectively expressed in astrocytes to confirm that astrocytic cAMP increases are coupled to glycogenolysis.

The data reported are interesting. The experiments are well conducted, the methodological approach is appropriate and the conclusions drawn by the authors are, in general, convincing. This study highlights a relevant aspect in the complexity of astrocyte signaling.

I have, however, a number of criticisms that need to be addressed by the authors.

1. A major criticism concerns the interpretation of the glycogen data. In the abstract the authors claim that "... repeated aversive stimuli lead to prolonged periods of vigilance accompanied by robust noradrenergic axonal activity and gradual sustained cAMP increases, triggering glycogenolysis". After reading this, one would expect a detailed study of the glycogenolysis in relation to the described repeated aversive stimuli. At the end of the introduction the authors also state that "... we demonstrate that astrocytic cAMP levels depend on the level of vigilance, with increases associated with prolonged phasic activity of NA/LC axons promoting glycogenolysis". Again, it seems that this study directly couples cAMP levels induced by vigilance to glycogenolysis. However, the only

experiment in which glycogen is measured employs Gs-type DREADD stimulation, which results in a huge increase of cAMP ($180.6 \pm 21.4 \%$), much higher than that that observed during the foot shock protocol, in which the maximal response was $108.3 \pm 2.3 \%$. This experiment confirms the already described role of cAMP in the promotion of energy metabolism (see for example Zhou Z. et al, 2019). To provide a clear support to the authors' conclusions, glycogen should be measured upon the behavioural tests performed in the study.

2. The last sentence of the Introduction "Together, these experiments clarify the integration of distinct NA signalling modes of astrocytes during learning" seems too pretentious, since no data are provided about the integration of the different signalling modes, but only on the differences in kinetics and threshold of the responses.

3. The analysis of calcium and cAMP signals has been restricted to astrocytic somata. The sensors that have been used by the authors would allow to reveal the dynamics of these signals at the level of the astrocytic processes, including the microdomains at the thin processes. Perhaps, data from these astrocyte compartments may tell another story....

4. On page 5, the authors describes a heterogeneity of the second messenger response among the astrocytes, suggesting a different expression ratio of the receptors. This observation is certainly interesting. However, more experiments appear necessary to better understand this issue. The authors dedicate to this an entire subheading in the Discussion, but they themselves recognized that the functional importance of their observations needs to be further investigated. I would suggest the authors to provide a further, more convincing evidence for the heterogeneity or tune down their interpretation of the data.

5. Further experiment and analysis are required regarding the air puff experiments described on Page 6. I would suggest to quantify also the frequency and the total time of MP signals, similarly to what is done for fear conditioning paradigm, to correlate these parameters to cAMP response. The increase in MP duration is, indeed, similar across these two paradigms. If NAergic activity is responsible for cAMP response, there should be a clear difference. Also, measuring NA content with dLight would help to clarify that this paradigm is similar to "short" PS episodes...

6. The authors mimic with optogenetics the pattern of NAergic activation induced by fear conditioning and measured the NA content by dLight signal. I can't understand why they did not measure the NA content directly during fear conditioning. This would be a more elegant and direct demonstration of the NA changes.

Minor Points:

Throughout the text, parietal and auditory cortex are often cited, sometimes in a confusing way. It is unclear whether and why a given experiment was performed in the auditory cortex, the parietal cortex or both.

In Fig. 1H, a pseudocolor scale should be added to the images in the panel (same point applies also to other similar images, e.g. dLight images). Furthermore, it is not clear whether the images are maximum projections or single focal plane images.

Fig. 1H-I: it should help to label the panels to indicate which PS duration they are referred to (H: 5s PS and I: 30s PS). Same thing applies to fig.2A.

The abbreviation PS should be explained in the legend of Fig. 1H&I and not Fig. 1J&K.

The final part of Fig. 1 legend is missing.

In Fig. 2 B to F, the same Y scale should be used.

In Fig. 3A, all the responses to repetitive PSs are shown, and it is not clear if the measure reported in the text at page 4 (i.e. $120 \pm 6.6\%$ for calcium) refers to the average response of astrocytes from all individual PS episodes or to the average peak amplitude of the response considered as a whole (i.e. the maximal response throughout PS episodes). All this should be more clearly stated.

Page 4, last row, refers to Fig. 3D for dLight data.

In Fig. 4 legend D, E. "... ... are composed of MP signals (arrowhead) and MP signals (yellow period). Please correct.

In Fig. 4J the graph seems incomplete, since the description is about MP duration...

In Fig. 5 legend of B,D,E: "Air puff given for 10s induces significant Ca²⁺ increases" and not "Air puff are given for..."

Fig. 4C Why a 5 s delay is interposed between GCaMP and RCaMP imaging? Is this referred only to the reported images?

Fig. 4J the graph seems incomplete, since the description is about MP duration...

Page 6, in the description of the fear conditioning paradigm, no explanation is provided about post FS1 and post FS2 periods, shown in Fig. 6B.

Fig. 6E and F . It is unclear the time point at which the images of post-FS1 and post-FS2 have been taken.

Reviewer #3:

Remarks to the Author:

In the manuscript by Oe and colleagues, the authors have put forth a tantalizing hypothesis that activation of the noradrenergic system, depending upon the behavior context, regulates distinct second messenger pathways in astrocytes. In this study, authors used state of the art genetically encoded Ca²⁺ (variants of GCaMPs and RCaMPs), cAMP (Pink Flamindo) and dopamine/norepinephrine (dLight) sensors. Using 2-photon microscopy, authors imaged simultaneous activation of Ca²⁺ and cAMP second messenger pathways in the cortical astrocytes of the awake and behaving mice. Based on their results, authors suggest that transient release of norepinephrine (NE) activates α 1 adrenergic receptor on astrocytes, which induce intracellular Ca²⁺ transients. In contrast, a prolonged release of NE activates the cAMP pathway in astrocytes, which authors suggest is caused by activation of β 2 adrenergic receptors. This study has a great potential to explore a novel aspect of cross-talk between Ca²⁺ and cAMP signaling in astrocytes, and the role of this mode of signaling in neural circuit function. However, several caveats reduce the overall enthusiasm for this study.

Major concerns:

1. Air puff mediated startle response leads to significant and transient Ca²⁺ response in cortical astrocytes (Fig. 5B, D, E). However, a similar robust stimulus doesn't induce an increase in cAMP (Fig. 5C, F, G) in astrocytes. Based on these results, the authors conclude that such startle based transient activation of the noradrenergic system doesn't activate cAMP pathways. It is unclear whether this finding is a 'real' biological phenomenon or an effect seen due to the low sensitivity of the cAMP sensor, Pink Flamindo, used in this study? A recent study (Harada et al., 2017) that designed and described this sensor clearly shows, to get a reasonable response from Pink Flamindo, a very high cytosolic concentration of cAMP is required, which was achieved by applying 100 μ M Forskolin. Also, in the current study, authors had to optogenetically photostimulated noradrenergic neurons in the locus coeruleus (LC) continuously for 30s to see any reliable increase (about 15%) in Pink Flamindo fluorescence in astrocytes (Fig. 1I, K and M; Fig. 2B).

2. In the current study, authors have used several state-of-the-art genetic encoded sensors. Still, many of these sensors are not optimized for detecting their corresponding molecules. The dopamine sensor, dLight, was used to study NE release (Fig. 2). dLight is 70 times less sensitive in detecting NE than dopamine, hence only very long bouts of NE release could be detected (Fig. 2F, G). Similar to Pink Flamindo, continuous optogenetic stimulation of LC neurons for 30s led to a reliable increase (about 15%) in dLight fluorescence. It is unclear, what is the physiological relevance of these long bouts of NE release and if such prolonged bouts occur frequently? Besides, the authors need to provide direct evidence that Ca²⁺ increase in axons (seen by GCaMPs) projecting from LC can be

directly co-related with the NE release from these axons (seen by dLight) (Fig. 4, Fig. 5H).

3. Authors don't provide experimental evidence that the cAMP elevations in response to either prolonged optogenetic activation of noradrenergic neurons in LC (Fig. 1) or during the fear-conditioning paradigm (Fig. 6F, H, J) are due to the direct activation of β 2-adrenergic receptors on astrocytes (SFig. 6). In the intact imaging preparations, such as one used in this study, a highly aversive stimulus like foot shock will engage almost all neurotransmitter and neuromodulator systems. Therefore, an extensive pharmacological dissection and expression analysis are essential to support a conclusion that the modest cAMP elevations seen in astrocytes are mainly due to direct activation of β 2-adrenergic receptors on these cells.

4. In Figure 3, authors suggest that there is a heterogenous astrocyte population in the auditory cortex, such as cells with high Ca^{2+} and cAMP response (active cells) and cells with low Ca^{2+} and high cAMP response (less active cells), and all other possible combinations. Here, authors don't provide experimental evidence that both Ca^{2+} and cAMP sensors were co-transfected in all astrocytes analyzed. The variability in the number of "active" and "less active" cells might quite well emerge from the differences in the fraction of cells virally transfected with one or the other sensor (Fig. 3L, M). Also, when cells were co-transfected with both Ca^{2+} and cAMP sensors, it will still be challenging to rule out the levels of expression of these sensors in each cell.

5. Astrocytes elevate both the second messengers, Ca^{2+} and cAMP, in response to the first foot sock, but during later foot socks these responses were diminished (Fig. 6, H). What is the mechanism of these diminished response seen in the fear conditioning paradigm, given that repeated startle can induce a robust Ca^{2+} response in astrocytes each time (Ding et al., 2013 and Paukert et al., 2014)? Also, it will be essential to see the direct dLight response, and hence NE releases, during the fear conditioning paradigm, instead of optogenetic stimulation of LC neurons shown in Fig. 7 N, O.

6. Although the entire study was performed in the auditory cortex, authors show enhanced glycogenolysis in response to activation of cAMP pathways, using Gs activating DREADDS, in the hippocampus (SFig. 5D, E). Is this phenomenon specific for hippocampal astrocytes, and is not seen in the cortical astrocytes?

Minor concerns:

- Fig. 1O – why no there no statistical significance between 1s and 2s PS, and 2s and 3s PS. Although Fig. 1L shows a clear difference during these time points.
- When comparing data, it will be helpful for the readers when the y-axis is plotted at the same scale i.e., starting from 0 (Fig. 1N-Q).
- On page 7, 4th line (from the end), either Fig. 4L is missing, or there is a typo, and it should be Fig. 7D, L.
- References 35, 51 need to be edited according to the journal specifications.

Reviewer #1

1- A critical aspect of the manuscript is based on the fact that cAMP signaling is typically triggered by a rather non-physiological manipulation, namely, 10 s photostimulation. What endogenous activity is this photostimulation mimicking? Under what physiological circumstances do astrocytes show similar activity as reported here by artificial activation? Such induction protocol raises the key question: In what physiological conditions does this phenomenon happen? Although the author showed input/output function to determine what kind of activity is required to triggers, the authors must show that action potentials can trigger these phenomena and to prove that it is not a technical artifact of the experimental approaches used in this study.

We agree completely with Reviewer #1 that increases of the astrocytic second messengers have to be validated under physiological conditions. For this reason, we expressed GCaMP in noradrenergic (NA) neurons in LC, as axonal Ca^{2+} activities reflect action potential (Forti et al., J Physiol. 2000), and performed simultaneous imaging of LC-NA axonal activity and astrocytic Ca^{2+} activity in the cortex (Fig. 4). We find that tonic action potential activity does not recruit astrocytic activation, while multi-peak bursting axonal activity reliably triggers astrocytic Ca^{2+} elevations. This situation is reproduced in the facial air-puff startle response, where multi-peak bursting axonal activity drives astrocytic Ca^{2+} activity, however, the brief occurrence of multi-peak axonal activity does not recruit cAMP activity (Fig. 5). This observation is consistent with the optogenetic experiment; in which we demonstrate the necessity of prolonged LC-NA axonal activity for astrocytic cAMP signaling (Fig. 1). Finally, we show that a train of multi-peak phasic activity is elicited during fear conditioning (Fig. 7), which is accompanied by astrocytic cAMP elevation (Fig. 6). Optogenetic stimulation of such LC-NA axonal activity led to accumulation of extracellular NA (Fig. 7N, O), which is consistent with the extracellular increase observed during fear conditioning (Supplementary Fig. 9). Collectively, we provide a comprehensive set of data that outline the physiological conditions for the activation of astrocytic Ca^{2+} and cAMP signaling in relation to LC-NA axonal activity. We believe that these added experiments fulfil the reviewer's concern.

2.- How does NA activate astrocytes? How does it work? What kinds of receptors are activated? In the abstract section the authors stated: “ α 1-adrenergic receptors triggers rapid astrocytic Ca^{2+} ” and “ β -adrenergic receptors elevates cAMP levels”. The authors should check these statements with experimental data as part of this study (for example, using pharmacological approaches).

Following this logical and constructive suggestion, we have dissected astrocytic second messenger dynamics by pharmacologically blocking adrenergic receptor subtypes. As a result, we found that Ca^{2+} and cAMP elevations are blocked by the alpha-1 receptor antagonist prazosin and the non-selective beta receptor antagonist propranolol, respectively. Further, we examined the contribution of beta-1 and beta-2 receptors using the respective antagonists betaxolol and ICI 118,551 (Fig. 1). Accordingly, we found a dominant contribution of beta-1 activation for the elevation of cAMP, which is consistent with the Brain RNA Seq mouse transcriptome database (Zhang et al. J Neurosci 2014). (lines 117-128)

3.- I am confused regarding the absence of dLight responses to low extracellular NA concentrations. In my view the paper would gain in clarity if they could show dLight doses-response curve, applying different known concentrations of NA and DA. Fig. 3B. What endogenous activity is this

photostimulation mimicking?

We are sorry to cause the confusion. Since the submission of the manuscript, the NA sensor nLight has become available through collaboration with Lin Tian (UC Davis, Supplementary Fig. 3). Using nLight, which has an order of magnitude higher affinity to NA over DA, we have re-examined extracellular NA. The new Figure 2 shows that nLight can detect low concentrations of extracellular NA after 3-s PS of LC-NA axons. As such, all dLight data were replaced with nLight data in this revision. PS is composed of 10 Hz pulse stimulation, which is meant to mimic phasic NAergic firing; Indeed, spaced presentation of PS at an interval of LC-NA phasic activity observed during fear conditioning resulted in an increase of NA level comparable to that of naïve mice after foot shock. (lines 129-147)

4.- Along the same line, does calcium signaling come from the cell soma, from the cell processes, or are they an average? Include details on how the ROIs were generated (both in neurons and in astrocytes). It is a key question after Volterra's lab work: Bindocci et al., Science 2017. See points 1 and 3.

All ROIs described in the main text were selected from individual astrocytes' somata. Following the reviewer's suggestion, we have added "All astrocyte responses are measured from the soma hereinafter unless otherwise noted." in Figure 1 legend. We describe this also in Methods under the Analysis subsection. We present our astrocyte process analyses in Supplementary Fig. 12. In this analysis, we demonstrate that process-wide cAMP signal is similar to the somatic signal, which is in line with the idea that cAMP elevation correlates with extracellular NA level. Moreover, we extended our analysis to investigate cAMP levels during periods of microdomain Ca²⁺ elevation. We found that transient microdomain Ca²⁺ events (<

20 s) do not accompany cAMP elevations, but long-lasting microdomain Ca^{2+} events (> 20 s) accompany mild cAMP elevations. We were not able to analyze neuronal soma, since we did not express Pink Flamindo in neurons in this study. (lines 359-372)

5.- Figure 4. What about cAMP in awake animals? What about astrocytic cAMP in basal states? Is there any correlation between NAergic multipeak Ca^{2+} signals and astrocytic cAMP signals? These results should be shown in the figure.

In a limited number of examples, we attempted to compare if the PinkFlamindo baseline signal changes between anesthesia and wakefulness, however, the results were not clear. This could be due to low baseline levels of cAMP or other confounding factors such as pH change during anesthesia. As for NAergic multipeak Ca^{2+} signal vs. astrocytic cAMP signal, we observed cAMP elevations during high vigilance states following aversive stimulus (fear conditioning, Fig. 6). Moreover, we show that NAergic multipeak Ca^{2+} signals occur continually during the high vigilance states (Fig. 7 A, H). Mimicking the multipeak signals by LC-NA axon optogenetic stimulation induced comparable increases of extracellular NA levels. Likewise, sparse activation of multipeak-like optogenetic LC-NA axon stimulation did not (Fig. 7 O, N). To better explain the cumulative increase of cortical NA levels by repeated phasic LC-NA activities, we prepared a schematic illustration in Supplementary Fig. 13.

6.- Page 6. The authors indicate “The next day presentation of the sound alone evoked high immobility index, verifying memory formation”. This is an important statement. However, I don't find the experimental data.

The data are shown in Fig. 6D. The graph indicates that sound presentation on the next day induces high immobility index in the absence of unconditional signal (foot shock). This conditioned response can be repeated at least three times, indicating that the fear memory has been consolidated. We have in response to the reviewer added reference to Fig. 6D (lines 256-257)

Minor points

7.- At times, the manuscript is difficult to read. It is useful explain the data and removed the percentage numbers.

Thank you for your constructive suggestion. We removed computation results which are not strictly pertinent to the conclusion of our manuscript or where the relative values are intuitively read from the plots. Where appropriate, we moved statistical values at the end of the sentence to keep the flow of the text.

Reviewer #2

1. A major criticism concerns the interpretation of the glycogen data. In the abstract the authors claim that "... repeated aversive stimuli lead to prolonged periods of vigilance accompanied by robust noradrenergic axonal activity and gradual sustained cAMP increases, triggering glycogenolysis". After reading this, one would expect a detailed study of the glycogenolysis in relation to the described repeated aversive stimuli. At the end of the introduction the authors also state that "... we demonstrate that astrocytic cAMP levels depend on the level of vigilance, with increases associated with prolonged phasic activity of NA/LC axons promoting glycogenolysis". Again, it seems that this study directly couples cAMP levels induced by vigilance to glycogenolysis. However, the only experiment in which glycogen is measured employs Gs-type DREADD stimulation, which results in a huge increase of cAMP (180.6 ± 21.4 %), much higher than that that observed during the foot shock protocol, in which the maximal response was 108.3 ± 2.3 %. This experiment confirms the already described role of cAMP in the promotion of energy metabolism (see for example Zhou Z. et al, 2019). To provide a clear support to the authors' conclusions, glycogen should be measured upon the behavioural tests performed in the study.

We thank for your logical and insightful comment. We have indeed considered performing microwave-assisted glycogen immunohistochemistry on head-restrained fear-conditioned mice, however, we find it technically difficult due to the high-power nature of focused microwave (FMW) irradiation: any non-biological attachment causes unwilling reactions (especially metals), and strongly discouraged by the manufacturer (Muromachi). We also considered removing the head plate after fear conditioning under anesthesia, however, this also proved difficult, as the headplate is strongly bonded to the skull. In our hands, an intact skull is a precondition for successful FMW irradiation, and even small damage to the skull has a destructive impact on the results. That said, we fully agree with this reviewer's criticism. In

the absence of direct demonstration, we agree that it is most reasonable to tone down on the link between fear memory-induced cAMP signaling and glycogen description.

2. The last sentence of the Introduction “Together, these experiments clarify the integration of distinct NA signalling modes of astrocytes during learning” seems too pretentious, since no data are provided about the integration of the different signalling modes, but only on the differences in kinetics and threshold of the responses.

We agree with the reviewer’s point about the overstatement. We have rephrased the sentence as follows:

Together, these experiments demonstrate distinct NA-induced signalling modes of astrocytes during learning. (lines 89-90)

3. The analysis of calcium and cAMP signals has been restricted to astrocytic somata. The sensors that have been used by the authors would allow to reveal the dynamics of these signals at the level of the astrocytic processes, including the microdomains at the thin processes. Perhaps, data from these astrocyte compartments may tell another story....

In the main text, we confined our analysis to somatic signals, we present our astrocyte process analyses in Supplementary Fig. 12. In this analysis, we demonstrate that process-wide cAMP signal is similar to somatic signal, which is in line with the idea that cAMP elevation correlate with extracellular NA level. Moreover, we extended our analysis to investigate cAMP levels during periods of microdomain Ca^{2+} elevation. We found that transient microdomain Ca^{2+} events (< 20 s) do not accompany cAMP elevations, but long-lasting Ca^{2+} activities (> 20 s) accompany mild cAMP elevations. We were not able to analyze neuronal soma, since we did not express Pink Flamingo in neurons in this study. (lines 359-372)

4. On page 5, the authors describes a heterogeneity of the second messenger response among the astrocytes, suggesting a different expression ratio of the receptors. This observation is certainly interesting. However, more experiments appear necessary to better understand this issue. The authors dedicate to this an entire subheading in the Discussion, but they themselves recognized that the functional importance of their observations needs to be further investigated. I would suggest the authors to provide a further, more convincing evidence for the heterogeneity or tune down their interpretation of the data.

Since heterogeneity of astrocytes is an emerging important topic, we thought it was a good idea to put a subheading in the discussion. However, as this reviewer suggests, we also realize that our data may not be sufficient to elucidate the full nature of astrocyte heterogeneity. Therefore, we decided to withdraw the subheading in the discussion while mentioning the second messenger response heterogeneity in relation to astrocyte heterogeneity in glycogen storage (Oe et al., 2016), which is functionally related to cAMP signaling. (lines 436-441)

5. Further experiment and analysis are required regarding the air puff experiments described on Page 6. I would suggest to quantify also the frequency and the total time of MP signals, similarly to what is done for fear conditioning paradigm, to correlate these parameters to cAMP response. The increase in MP duration is, indeed, similar across these two paradigms. If NAergic activity is responsible for cAMP response, there should be a clear difference. Also, measuring NA content with dLight would help to clarify that this paradigm is similar to “short” PS episodes...

We re-analyzed the frequency and total time of MP signals to compare startle and fear conditioning. Analysis with 10-s and 30-s bins were performed and displayed in Supplementary Fig. 8. Both startle and fear conditioning responses showed high frequency

and total time of MP signals immediately after stimulation, however, the occurrence of MPs were more frequent and sustained after unconditional stimulus (foot shock) in fear conditioning, compared to air-puff-induced startle. (Gray lines represent the mean of control.) Due to a time and resource limitation, we were not able to measure d/nLight signals during whisker air puff. (lines 295-297)

6. The authors mimick with optogenetics the pattern of NAergic activation induced by fear conditioning and measured the NA content by dLight signal. I can't understand why they did not measure the NA content directly during fear conditioning. This would be a more elegant and direct demonstration of the NA changes.

We examined extracellular NA content with nLight which is an optimized probe for NA (from Lin Tian's lab, UC Davis, Supplementary Fig. 3). Extracellular NA was measured during fear conditioning from the 1st to 5th session, which demonstrated that early conditionings increase NA release, but the level of increase gradually attenuates in later sessions (Supplementary Fig 9). (lines 306-308)

Minor Points:

Throughout the text, parietal and auditory cortex are often cited, sometimes in a confusing way. It is unclear whether and why a given experiment was performed in the auditory cortex, the parietal cortex or both.

All experiments with awake mice were performed in the auditory cortex, as the conditional stimulus is a chirp sound. Experiments involving optogenetic activation of LC-NA axons were performed in the parietal cortex. Given the proof-of-principle nature of the latter, accessibility of epi-illumination for optogenetic stimulation (i.e. the parietal cortex is more flat, which provides even illumination), and the diffuse manner of LC-NA axonic projection, I sincerely hope that the experiments were acceptable. To avoid confusion, we stated the following in the methods section: “All experiments with awake mice were performed in the auditory cortex, whereas experiments involving optogenetic activation of LC-NA axons were performed in the parietal cortex.” (lines 650-651)

In Fig. 1H, a pseudocolor scale should be added to the images in the panel (same point applies also to other similar images, e.g. dLight images). Furthermore, it is not clear whether the images are maximum projections or single focal plane images.

We put calibration bars for all pseudocolor images including nLight images. Most analyses were performed with single focal plane images. However, long-time imaging experiments involving DREADD activation were performed with 10 μm maximum projections to reduce the effect of z-axis (depth) drift, which sometimes occurs in during the course of 10 min or so.

Fig. 1H-I: it should help to label the panels to indicate which PS duration they are referred to (H: 5s PS and I: 30s PS). Same thing applies to fig.2A.

Thank you for your suggestion. We labeled individual PS duration.

The abbreviation PS should be explained in the legend of Fig. 1H&I and not Fig. 1J&K.

Thank you for your suggestion. Because of limited space in the figure, we put the abbreviation PS in the figure legend.

The final part of Fig. 1 legend is missing.

Thank you for correction of our mistake. We put complete version of legend in Fig1.

In Fig. 2 B to F, the same Y scale should be used.

Thank you for your suggestion. We applied same Y scale for new Fig2.

In Fig. 3A, all the responses to repetitive PSs are shown, and it is not clear if the measure reported in the text at page 4 (i.e. $120 \pm 6.6\%$ for calcium) refers to the average response of astrocytes from all individual PS episodes or to the average peak amplitude of the response considered as a whole (i.e. the maximal response throughout PS episodes). All this should be more clearly stated.

We are sorry for the confusion. Basically, we analyzed average of individual sessions for most of data. However, since we thought detailed and careful analysis in the beginning is needed, we analyzed individual cell responses in Fig1. We stated this in the figure legends.

Page 4, last row, refers to Fig. 3D for dLight data.

Thank you for correction. We removed d/nLight from the Fig. 3D.

In Fig. 4 legend D, E. "... ... are composed of MP signals (arrowhead) and MP signals (yellow period). Please correct.

Thank you for correction. We changed the legend accordingly.
"SP signals (arrowhead) and MP signals (yellow period)"

In Fig. 4J the graph seems incomplete, since the description is about MP duration...

Thank you for your suggestion. J is for single peak (SP) duration, therefore we change y-label accordingly.

In Fig. 5 legend of B,D,E: "Air puff given for 10s induces significant Ca^{2+} increases" and not "Air puff are given for..."

Thank you for your suggestion. We corrected the sentence accordingly.

Fig. 4C Why a 5 s delay is interposed between GCaMP and RCaMP imaging? Is this referred only to the reported images?

I'm sorry with our imcomplete description. NAergic MP signals start slightly prior to astrocytic Ca^{2+} , therefore we took Ca^{2+} images with 5 s delay.

Page 6, in the description of the fear conditioning paradigm, no explanation is provided about post FS1 and post FS2 periods, shown in Fig. 6B.

We appreciate reviewer's careful correction. We added explanation about post-FS1, 2 in the main text (Lines 247–248).

Fig. 6E and F . It is unclear the time point at which the images of post-FS1 and post-FS2 have been taken.

Thank you for your suggestion. We stated the timing in figure legend.

Reviewer #3

1. Air puff mediated startle response leads to significant and transient Ca^{2+} response in cortical astrocytes (Fig. 5B, D, E). However, a similar robust stimulus doesn't induce an increase in cAMP (Fig. 5C, F, G) in astrocytes. Based on these results, the authors conclude that such startle based transient activation of the noradrenergic system doesn't activate cAMP pathways. It is unclear whether

this finding is a ‘real’ biological phenomenon or an effect seen due to the low sensitivity of the cAMP sensor, Pink Flamindo, used in this study? A recent study (Harada et al., 2017) that designed and described this sensor clearly shows, to get a reasonable response from Pink Flamindo, a very high cytosolic concentration of cAMP is required, which was achieved by applying 100 μ M Forskolin. Also, in the current study, authors had to optogenetically photostimulated noradrenergic neurons in the locus coeruleus (LC) continuously for 30s to see any reliable increase (about 15%) in Pink Flamindo fluorescence in astrocytes (Fig. 1I, K and M; Fig. 2B).

Our previous study (Harada et al., 2017) demonstrated that Pink Flamindo can detect low concentrations of cAMP (~200nM), and we show here that a decrease of cAMP from the base line is detectable after Gi pathway activation by hM4D DREADD (Supplementary Fig. 10). Considering that basal cAMP levels of mouse N1E-115 neuroblastoma cells are $\sim 0.4 \pm 0.3 \mu$ M and can increase up to 9 μ M (Salonikidis, 2008), the demonstration of both increase and decrease of Pink Flamindo signal warrants that our *in vivo* measurements of cAMP by this probe reflect physiological changes. In response to this reviewer’s concern, we added the text in the discussion (lines 334-340).

2. In the current study, authors have used several state-of-the-art genetical encoded sensors. Still, many of these sensors are not optimized for detecting their corresponding molecules. The dopamine sensor, dLight, was used to study NE release (Fig. 2). dLight is 70 times less sensitive in detecting NE than dopamine, hence only very long bouts of NE release could be detected (Fig. 2F, G). Similar to Pink Flamindo, continuous optogenetic stimulation of LC neurons for 30s led to a reliable increase (about 15%) in dLight fluorescence. It is unclear, what is the physiological relevance of these long bouts of NE release and if such prolonged bouts occur frequently? Besides, the authors need to provide direct evidence that Ca²⁺ increase in axons (seen by GCaMPs) projecting from LC can be directly co-related with the NE release from these axons (seen by dLight) (Fig. 4, Fig. 5H).

This reviewer’s criticisms are very reasonable, in particular, the one about the usage of dLight to detect NA instead of DA. Indeed, other reviewers have expressed similar concerns. To this end, we re-examined extracellular NA by using the NA-optimized nLight (Patriarchi et al., 2018; collaboration with Lin Tian, UC Davis, Supplementary Fig. 3). We were pleased to see that this probe can detect lower NA levels that were not detectable by dLight. In fact, we replaced dLight measurements with nLight in Figure 2 and elsewhere. As in our answer to comment #1, we believe that the DREADD-Gi experiments provide convincing support for the detectability of physiological cAMP signals by Pink Flamindo. In the beginning of this study (Figs.1-3), we focused on exploring detailed properties of astrocytic second messengers which found astrocytic Ca²⁺ and cAMP has temporally distinct dynamics and such differences

are most possibly caused by differential affinity of alpha-1 and beta receptors. However, as this reviewer mentioned, there is a concern that some of the stimuli in Figs. 1-3 might not reflect a physiological situation. Therefore, we sought to examine physiological conditions that triggers detectable cAMP elevations in Figs. 4-7. These results indicate that repeated NAergic multippeak signals induces cAMP increases in physiological conditions.

3. Authors don't provide experimental evidence that the cAMP elevations in response to either prolonged optogenetic activation of noradrenergic neurons in LC (Fig. 1) or during the fear-conditioning paradigm (Fig. 6F, H, J) are due to the direct activation of β_2 -adrenergic receptors on astrocytes (SFig. 6). In the intact imaging preparations, such as one used in this study, a highly aversive stimulus like foot shock will engage almost all neurotransmitter and neuromodulator systems. Therefore, an extensive pharmacological dissection and expression analysis are essential to support a conclusion that the modest cAMP elevations seen in astrocytes are mainly due to direct activation of β_2 -adrenergic receptors on these cells.

Following the suggestion of pharmacological dissection of LC-NA transmission-induced signaling in cortical astrocytes by this and Reviewer #1, we examined performed experiments using the alpha-1 antagonist prazosin, non-selective beta antagonist propranolol, beta-1 antagonist betaxolol and beta-2 antagonist ICI 118,551, which we present in Fig. 1. We find that astrocytic Ca^{2+} was blocked by alpha-1 antagonism, while astrocytic cAMP was blocked by beta receptors, particularly the beta-1 receptor. The dominant role of the beta-1 over beta2 receptor is consistent with the Brain RNA Seq mouse transcriptome database (Zhang et al. J Neurosci 2014). We hope that the pharmacological analysis in combination with citing this well-established transcriptomic literature would suffice the concern of this reviewer. Additionally, we confirmed that NA is responsible for astrocytic Ca^{2+} and cAMP increases

during fear conditioning by pharmacological dissection (prazosin + propranolol) in Supplementary Fig. 7. (lines 270-272)

4. In Figure 3, authors suggest that there is a heterogenous astrocyte population in the auditory cortex, such as cells with high Ca^{2+} and cAMP response (active cells) and cells with low Ca^{2+} and high cAMP response (less active cells), and all other possible combinations. Here, authors don't provide experimental evidence that both Ca^{2+} and cAMP sensors were co-transfected in all astrocytes analyzed. The variability in the number of "active" and "less active" cells might quite well emerge from the differences in the fraction of cells virally transfected with one or the other sensor (Fig. 3L, M). Also, when cells were co-transfected with both Ca^{2+} and cAMP sensors, it will still be challenging to rule out the levels of expression of these sensors in each cell.

The reviewer has raised a reasonable concern that the proposed heterogeneity might have been caused by varying expression of GCaMP and Pink Flamindo. We have addressed this concern in Supplementary Fig. 4. To examine further on this issue, we plotted basal GCaMP and PinkFlamindo fluorescence signals for each astrocyte. We find that all cells that were subject to the analysis co-expressed the probes and the level of expression was highly correlated. This result supports that the observation of Ca^{2+} -response-prone or cAMP-response-prone cells are not due to the biased expression of respective probes. Next, we examined if basal fluorescence intensity influences the degree of relative signal changes ($\Delta F/F$). We find that neither GCaMP nor Pink Flamindo has significant correlation between basal fluorescence level and relative signal change within the set of cells that we analyzed in this study (lines 191-200).

5. Astrocytes elevate both the second messengers, Ca^{2+} and cAMP, in response to the first foot sock, but during later foot socks these responses were diminished (Fig. 6, H). What is the mechanism of these diminished response seen in the fear conditioning paradigm, given that repeated startle can induce a robust Ca^{2+} response in astrocytes each time (Ding et al., 2013 and Paukert et al., 2014)? Also, it will be essential to see the direct dLight response, and hence NE releases, during the fear conditioning paradigm, instead of optogenetic stimulation of LC neurons shown in Fig. 7 N, O.

We would think that novelty is an important factor for the reduction of Ca^{2+} and cAMP. Startle stimulation is given abruptly, and this may be taken as “unexpected, novelty” signal to transiently drive the noradrenergic signal. On the other hand, fear conditioning gives sound as a sign for shortly coming foot shock, therefore animals can learn and expect foot shock, therefore foot shock becomes less novel with repeated conditioning. According to this reviewer’s suggestion, we measured NA level during fear conditioning with the hypothesis that NA release correlates with novelty. As a result, NA release increased in early sessions and gradually disappeared in later sessions (Supplementary Fig. 9). Together with the pharmacological blockade experiments during fear conditioning (Supplementary Fig. 7), our results suggest that released amount of NA determines the Ca^{2+} and cAMP responses after conditional stimulus (foot shock). (lines 270-272; 306-308)

6. Although the entire study was performed in the auditory cortex, authors show enhanced glycogenolysis in response to activation of cAMP pathways, using Gs activating DREADDS, in the hippocampus (SFig. 5D, E). Is this phenomenon specific for hippocampal astrocytes, and is not seen in the cortical astrocytes?

We added new cortical glycogen immunohistochemical micrographs with Gs activation by DREADD. Similar to our original observation in the hippocampus, cortical glycogen also diminished after the activation of Gs signaling pathway (Fig. 8). (lines 309-319)

F Gs-DREADD in the cerebral cortex

Minor concerns:

- Fig. 1O – why no there no statistical significance between 1s and 2s PS, and 2s and 3s PS. Although Fig. 1L shows a clear difference during these time points.

Whiskers in a box plot is sometimes deceiving as they are often confused with error bars that represent the standard deviation or standard error of the mean. We used a standard box plot, where the length of the whisker is $1.5 \times (Q3-Q1)$, Q3 and Q1 denote quartile (25 and 75 percentile) values.

- When comparing data, it will be helpful for the readers when the y-axis is plotted at the same scale i.e., starting from 0 (Fig. 1N-Q).

Different fluorescent probes have distinct dynamic ranges, therefore it's difficult to apply same scale.

- On page 7, 4th line (from the end), either Fig. 4L is missing, or there is a typo, and it should be Fig. 7D, L.

The typos are now corrected.

- References 35, 51 need to be edited according to the journal specifications.

The references were updated with the correct volume and page numbers.

Reviewers' Comments:

Reviewer #1:

Remarks to the Author:

The authors have satisfactorily addressed all my concerns. I consider the manuscript is suitable for publication in its current form.

Reviewer #2:

Remarks to the Author:

In their revised work, Dr Hirase and colleagues responded to my concerns in a direct and satisfactory manner. The new analyses of their original data as well as the results from the new experiments that they performed significantly strengthen the conclusions and the overall quality of this manuscript.

Reviewer #3:

Remarks to the Author:

The authors addressed all of my concerns. This manuscript has improved significantly and would contribute considerably to the field of astrocyte biology. I recommend the publication of this work.

REVIEWERS' COMMENTS:

Reviewer #1 (Remarks to the Author):

The authors have satisfactorily addressed all my concerns. I consider the manuscript is suitable for publication in its current form.

We appreciate reviewer's constructive comments.

Reviewer #2 (Remarks to the Author):

In their revised work, Dr Hirase and colleagues responded to my concerns in a direct and satisfactory manner. The new analyses of their original data as well as the results from the new experiments that they performed significantly strengthen the conclusions and the overall quality of this manuscript.

We appreciate reviewer's constructive comments.

Reviewer #3 (Remarks to the Author):

The authors addressed all of my concerns. This manuscript has improved significantly and would contribute considerably to the field of astrocyte biology. I recommend the publication of this work.

We appreciate reviewer's constructive comments.

Reviewer #1

1- A critical aspect of the manuscript is based on the fact that cAMP signaling is typically triggered by a rather non-physiological manipulation, namely, 10 s photostimulation. What endogenous activity is this photostimulation mimicking? Under what physiological circumstances do astrocytes show similar activity as reported here by artificial activation? Such induction protocol raises the key question: In what physiological conditions does this phenomenon happen? Although the author showed input/output function to determine what kind of activity is required to triggers, the authors must show that action potentials can trigger these phenomena and to prove that it is not a technical artifact of the experimental approaches used in this study.

We agree completely with Reviewer #1 that increases of the astrocytic second messengers have to be validated under physiological conditions. For this reason, we expressed GCaMP in noradrenergic (NA) neurons in LC, as axonal Ca²⁺ activities reflect action potential (Forti et al., J Physiol. 2000), and performed simultaneous imaging of LC-NA axonal activity and astrocytic Ca²⁺ activity in the cortex (Fig. 4). We find that tonic action potential activity does not recruit astrocytic activation, while multi-peak bursting axonal activity reliably triggers astrocytic Ca²⁺ elevations. This situation is reproduced in the facial air-puff startle response, where multi-peak bursting axonal activity drives astrocytic Ca²⁺ activity, however, the brief occurrence of multi-peak axonal activity does not recruit cAMP activity (Fig. 5). This observation is consistent with the optogenetic experiment; in which we demonstrate the necessity of prolonged LC-NA axonal activity for astrocytic cAMP signaling (Fig. 1). Finally, we show that a train of multi-peak phasic activity is elicited during fear conditioning (Fig. 7), which is accompanied by astrocytic cAMP elevation (Fig. 6). Optogenetic stimulation of such LC-NA axonal activity led to accumulation of extracellular NA (Fig. 7N, O), which is consistent with the extracellular increase observed during fear conditioning (Supplementary Fig. 9). Collectively, we provide a comprehensive set of data that outline the physiological conditions for the activation of astrocytic Ca²⁺ and cAMP signaling in relation to LC-NA axonal activity. We believe that these added experiments fulfil the reviewer's concern.

2.- How does NA activate astrocytes? How does it work? What kinds of receptors are activated? In the abstract section the authors stated: “ α 1-adrenergic receptors triggers rapid astrocytic Ca²⁺”

and “ β -adrenergic receptors elevates cAMP levels”. The authors should check these statements with experimental data as part of this study (for example, using pharmacological approaches).

Following this logical and constructive suggestion, we have dissected astrocytic second messenger dynamics by pharmacologically blocking adrenergic receptor subtypes. As a result, we found that Ca^{2+} and cAMP elevations are blocked by the alpha-1 receptor antagonist prazosin and the non-selective beta receptor antagonist propranolol, respectively. Further, we examined the contribution of beta-1 and beta-2 receptors using the respective antagonists betaxolol and ICI 118,551 (Fig. 1). Accordingly, we found a dominant contribution of beta-1 activation for the elevation of cAMP, which is consistent with the Brain RNA Seq mouse transcriptome database (Zhang et al. J Neurosci 2014). (lines 117-128)

3.- I am confused regarding the absence of dLight responses to low extracellular NA concentrations. In my view the paper would gain in clarity if they could show dLight doses-response curve, applying different known concentrations of NA and DA. Fig. 3B. What endogenous activity is this photostimulation mimicking?

We are sorry to cause the confusion. Since the submission of the manuscript, the NA sensor nLight has become available through collaboration with Lin Tian (UC Davis, Supplementary Fig. 3). Using nLight, which has an order of magnitude higher affinity to NA over DA, we have re-examined extracellular NA. The new Figure 2 shows that nLight can detect low concentrations of extracellular NA after 3-s PS of LC-NA axons. As such, all dLight data were replaced with nLight data in this revision. PS is composed of 10 Hz pulse stimulation, which is meant to mimic phasic NAergic firing; indeed, spaced presentation of PS at an interval of LC-NA phasic activity observed during fear conditioning resulted in an increase of NA level comparable to that of naïve mice after foot shock. (lines 129-147)

4.- Along the same line, does calcium signaling come from the cell soma, from the cell processes, or are they an average? Include details on how the ROIs were generated (both in neurons and in astrocytes). It is a key question after Volterra's lab work: Bindocci et al., Science 2017. See points 1 and 3.

All ROIs described in the main text were selected from individual astrocytes' somata. Following the reviewer's suggestion, we have added "All astrocyte responses are measured from the soma hereinafter unless otherwise noted." in Figure 1 legend. We describe this also in Methods under the Analysis subsection. We present our astrocyte process analyses in Supplementary Fig. 12. In this analysis, we demonstrate that process-wide cAMP signal is similar to the somatic signal, which is in line with the idea that cAMP elevation correlates with extracellular NA level. Moreover, we extended our analysis to investigate cAMP levels during periods of microdomain Ca²⁺ elevation. We

found that transient microdomain Ca^{2+} events (< 20 s) do not accompany cAMP elevations, but long-lasting microdomain Ca^{2+} events (> 20 s) accompany mild cAMP elevations. We were not able to analyze neuronal soma, since we did not express Pink Flamindo in neurons in this study. (lines 359-372)

5.- Figure 4. What about cAMP in awake animals? What about astrocytic cAMP in basal states? Is there any correlation between NAergic multipeak Ca^{2+} signals and astrocytic cAMP signals? These results should be shown in the figure.

In a limited number of examples, we attempted to compare if the PinkFlamindo baseline signal changes between anesthesia and wakefulness, however, the results were not clear. This could be due to low baseline levels of cAMP or other confounding factors such as pH change during anesthesia. As for NAergic multipeak Ca^{2+} signal vs. astrocytic cAMP signal, we observed cAMP elevations during high vigilance states

following aversive stimulus (fear conditioning, Fig. 6). Moreover, we show that NAergic multipeak Ca^{2+} signals occur continually during the high vigilance states (Fig. 7 A, H). Mimicking the multipeak signals by LC-NA axon optogenetic stimulation induced comparable increases of extracellular NA levels. Likewise, sparse activation of multipeak-like optogenetic LC-NA axon stimulation did not (Fig. 7 O, N). To better explain the cumulative increase of cortical NA levels by repeated phasic LC-NA activities, we prepared a schematic illustration in Supplementary Fig. 13.

6.- Page 6. The authors indicate “The next day presentation of the sound alone evoked high immobility index, verifying memory formation”. This is an important statement. However, I don't find the experimental data.

The data are shown in Fig. 6D. The graph indicates that sound presentation on the next day induces high immobility index in the absence of unconditional signal (foot shock). This conditioned response can be repeated at least three times, indicating that the fear memory has been consolidated. We have in response to the reviewer added reference

to Fig. 6D (lines 256-257)

Minor points

7.- At times, the manuscript is difficult to read. It is useful explain the data and removed the percentage numbers.

Thank you for your constructive suggestion. We removed computation results which are not strictly pertinent to the conclusion of our manuscript or where the relative values are intuitively read from the plots. Where appropriate, we moved statistical values at the end of the sentence to keep the flow of the text.

Reviewer #2

1. A major criticism concerns the interpretation of the glycogen data. In the abstract the authors claim that "... repeated aversive stimuli lead to prolonged periods of vigilance accompanied by robust noradrenergic axonal activity and gradual sustained cAMP increases, triggering glycogenolysis". After reading this, one would expect a detailed study of the glycogenolysis in relation to the described repeated aversive stimuli. At the end of the introduction the authors also state that "... we demonstrate that astrocytic cAMP levels depend on the level of vigilance, with increases associated with prolonged phasic activity of NA/LC axons promoting

glycogenolysis". Again, it seems that this study directly couples cAMP levels induced by vigilance to glycogenolysis. However, the only experiment in which glycogen is measured employs Gs-type DREADD stimulation, which results in a huge increase of cAMP (180.6 ± 21.4 %), much higher than that that observed during the foot shock protocol, in which the maximal response was 108.3 ± 2.3 %. This experiment confirms the already described role of cAMP in the promotion of energy metabolism (see for example Zhou Z. et al, 2019). To provide a clear support to the authors' conclusions, glycogen should be measured upon the behavioural tests performed in the study.

We thank for your logical and insightful comment. We have indeed considered performing microwave-assisted glycogen immunohistochemistry on head-restrained fear-conditioned mice, however, we find it technically difficult due to the high-power nature of focused microwave (FMW) irradiation: any non-biological attachment causes unwilling reactions (especially metals), and strongly discouraged by the manufacturer (Muromachi). We also considered removing the head plate after fear conditioning under anesthesia, however, this also proved difficult, as the headplate is strongly bonded to the skull. In our hands, an intact skull is a precondition for successful FMW irradiation, and even small damage to the skull has a destructive impact on the results. That said, we fully agree with this reviewer's criticism. In the absence of direct demonstration, we agree that it is most reasonable to tone down on the link between fear memory-induced cAMP signaling and glycogen description.

2. The last sentence of the Introduction "Together, these experiments clarify the integration of distinct NA signalling modes of astrocytes during learning" seems too pretentious, since no data are provided about the integration of the different signalling modes, but only on the differences in kinetics and threshold of the responses.

We agree with the reviewer's point about the overstatement. We have rephrased the sentence as follows:

Together, these experiments demonstrate distinct NA-induced signalling modes of astrocytes during learning. (lines 89-90)

3. The analysis of calcium and cAMP signals has been restricted to astrocytic somata. The sensors that have been used by the authors would allow to reveal the dynamics of these signals at the level of the astrocytic processes, including the microdome at the thin processes.

Perhaps, data from these astrocyte compartments may tell another story....

In the main text, we confined our analysis to somatic signals, we present our astrocyte process analyses in Supplementary Fig. 12. In this analysis, we demonstrate that process-wide cAMP signal is similar to somatic signal, which is in line with the idea that cAMP elevation correlate with extracellular NA level. Moreover, we extended our analysis to investigate cAMP levels during periods of microdomain Ca^{2+} elevation. We found that transient microdomain Ca^{2+} events (< 20 s) do not accompany cAMP elevations, but long-lasting Ca^{2+} activities (> 20 s) accompany mild cAMP elevations. We were not able to analyze neuronal soma, since we did not express Pink Flamindo in neurons in this study. (lines 359-372)

4. On page 5, the authors describes a heterogeneity of the second messenger response among the astrocytes, suggesting a different expression ratio of the receptors. This observation is certainly

interesting. However, more experiments appear necessary to better understand this issue. The authors dedicate to this an entire subheading in the Discussion, but they themselves recognized that the functional importance of their observations needs to be further investigated. I would suggest the authors to provide a further, more convincing evidence for the heterogeneity or tune down their interpretation of the data.

Since heterogeneity of astrocytes is an emerging important topic, we thought it was a good idea to put a subheading in the discussion. However, as this reviewer suggests, we also realize that our data may not be sufficient to elucidate the full nature of astrocyte heterogeneity. Therefore, we decided to withdraw the subheading in the discussion while mentioning the second messenger response heterogeneity in relation to astrocyte heterogeneity in glycogen storage (Oe et al., 2016), which is functionally related to cAMP signaling. (lines 436-441)

5. Further experiment and analysis are required regarding the air puff experiments described on Page 6. I would suggest to quantify also the frequency and the total time of MP signals, similarly to what is done for fear conditioning paradigm, to correlate these parameters to cAMP response. The increase in MP duration is, indeed, similar across these two paradigms. If NAergic activity is responsible for cAMP response, there should be a clear difference. Also, measuring NA content with dLight would help to clarify that this paradigm is similar to “short” PS episodes...

We re-analyzed the frequency and total time of MP signals to compare startle and fear conditioning. Analysis with 10-s and 30-s bins were performed and displayed in Supplementary Fig. 8. Both startle and fear conditioning responses showed high frequency and total time of MP signals immediately after stimulation, however, the occurrence of MPs were more frequent and sustained after unconditional stimulus (foot shock) in fear conditioning, compared to air-puff-induced startle. (Gray lines represent the mean of control.) Due to a time and resource limitation, we were not able to measure d/nLight signals during whisker air puff. (lines 295-297)

6. The authors mimic with optogenetics the pattern of NAergic activation induced by fear conditioning and measured the NA content by dLight signal. I can't understand why they did not measure the NA content directly during fear conditioning. This would be a more elegant and direct demonstration of the NA changes.

We examined extracellular NA content with nLight which is an optimized probe for NA (from Lin Tian's lab, UC Davis, Supplementary Fig. 3). Extracellular NA was measured during fear conditioning from the 1st to 5th session, which demonstrated that early conditionings increase NA release, but the level of increase gradually attenuates in later sessions (Supplementary Fig 9). (lines 306-308)

Minor Points:

Throughout the text, parietal and auditory cortex are often cited, sometimes in a confusing way. It is unclear whether and why a given experiment was performed in the auditory cortex, the parietal cortex or both.

All experiments with awake mice were performed in the auditory cortex, as the conditional stimulus is a chirp sound. Experiments involving optogenetic activation of LC-NA axons were performed in the parietal cortex. Given the proof-of-principle nature of the latter, accessibility of epi-illumination for optogenetic stimulation (i.e. the parietal cortex is more flat, which provides even illumination), and the diffuse manner of LC-NA axonic projection, I sincerely hope that the experiments were acceptable. To avoid confusion, we stated the following in the methods section: “All experiments with awake mice were performed in the auditory cortex, whereas experiments involving optogenetic activation of LC-NA axons were performed in the parietal cortex.” (lines 650-651)

In Fig. 1H, a pseudocolor scale should be added to the images in the panel (same point applies also to other similar images, e.g. dLight images). Furthermore, it is not clear whether the images are maximum projections or single focal plane images.

We put calibration bars for all pseudocolor images including nLight images. Most analyses were performed with single focal plane images. However, long-time imaging experiments involving DREADD activation were performed with 10 μm maximum

projections to reduce the effect of z-axis (depth) drift, which sometimes occurs in during the course of 10 min or so.

Fig. 1H-I: it should help to label the panels to indicate which PS duration they are referred to (H: 5s PS and I: 30s PS). Same thing applies to fig.2A.

Thank you for your suggestion. We labeled individual PS duration.

The abbreviation PS should be explained in the legend of Fig. 1H&I and not Fig. 1J&K.

Thank you for your suggestion. Because of limited space in the figure, we put the abbreviation PS in the figure legend.

The final part of Fig. 1 legend is missing.

Thank you for correction of our mistake. We put complete version of legend in Fig1.

In Fig. 2 B to F, the same Y scale should be used.

Thank you for your suggestion. We applied same Y scale for new Fig2.

In Fig. 3A, all the responses to repetitive PSs are shown, and it is not clear if the measure reported in the text at page 4 (i.e. $120 \pm 6.6\%$ for calcium) refers to the average response of astrocytes from all individual PS episodes or to the average peak amplitude of the response considered as a whole (i.e. the maximal response throughout PS episodes). All this should be more clearly stated.

We are sorry for the confusion. Basically, we analyzed average of individual sessions for most of data. However, since we thought detailed and careful analysis in the beginning is needed, we analyzed individual cell responses in Fig1. We stated this in the figure legends.

Page 4, last row, refers to Fig. 3D for dLight data.

Thank you for correction. We removed d/nLight from the Fig. 3D.

In Fig. 4 legend D, E. "... ... are composed of MP signals (arrowhead) and MP signals (yellow period). Please correct.

Thank you for correction. We changed the legend accordingly.

“SP signals (arrowhead) and MP signals (yellow period)”

In Fig. 4J the graph seems incomplete, since the description is about MP duration...

Thank you for your suggestion. J is for single peak (SP) duration, therefore we change y-label accordingly.

In Fig. 5 legend of B,D,E: “Air puff given for 10s induces significant Ca²⁺ increases” and not “Air puff are given for...”

Thank you for your suggestion. We corrected the sentence accordingly.

Fig. 4C Why a 5 s delay is interposed between GCaMP and RCaMP imaging? Is this referred only to the reported images?

I'm sorry with our incomplete description. NAergic MP signals start slightly prior to astrocytic Ca²⁺, therefore we took Ca²⁺ images with 5 s delay.

Page 6, in the description of the fear conditioning paradigm, no explanation is provided about post FS1 and post FS2 periods, shown in Fig. 6B.

We appreciate reviewer's careful correction. We added explanation about post-FS1, 2 in the main text (Lines 247–248).

Fig. 6E and F . It is unclear the time point at which the images of post-FS1 and post-FS2 have been taken.

Thank you for your suggestion. We stated the timing in figure legend.

Reviewer #3

1. Air puff mediated startle response leads to significant and transient Ca²⁺ response in cortical astrocytes (Fig. 5B, D, E). However, a similar robust stimulus doesn't induce an increase in

cAMP (Fig. 5C, F, G) in astrocytes. Based on these results, the authors conclude that such startle based transient activation of the noradrenergic system doesn't activate cAMP pathways. It is unclear whether this finding is a 'real' biological phenomenon or an effect seen due to the low sensitivity of the cAMP sensor, Pink Flamindo, used in this study? A recent study (Harada et al., 2017) that designed and described this sensor clearly shows, to get a reasonable response from Pink Flamindo, a very high cytosolic concentration of cAMP is required, which was achieved by applying 100 μ M Forskolin. Also, in the current study, authors had to optogenetically photostimulated noradrenergic neurons in the locus coeruleus (LC) continuously for 30s to see any reliable increase (about 15%) in Pink Flamindo fluorescence in astrocytes (Fig. 1I, K and M; Fig. 2B).

Our previous study (Harada et al., 2017) demonstrated that Pink Flamindo can detect low concentrations of cAMP (~200nM), and we show here that a decrease of cAMP from the base line is detectable after Gi pathway activation by hM4D DREADD (Supplementary Fig. 10). Considering that basal cAMP levels of mouse N1E-115 neuroblastoma cells are $\sim 0.4 \pm 0.3 \mu$ M and can increase up to 9 μ M (Salonikidis, 2008), the demonstration of both increase and decrease of Pink Flamindo signal warrants that our *in vivo* measurements of cAMP by this probe reflect physiological changes. In response to this reviewer's concern, we added the text in the discussion (lines 334-340).

2. In the current study, authors have used several state-of-the-art genetical encoded sensors. Still,

many of these sensors are not optimized for detecting their corresponding molecules. The dopamine sensor, dLight, was used to study NE release (Fig. 2). dLight is 70 times less sensitive in detecting NE than dopamine, hence only very long bouts of NE release could be detected (Fig. 2F, G). Similar to Pink Flamindo, continuous optogenetic stimulation of LC neurons for 30s led to a reliable increase (about 15%) in dLight fluorescence. It is unclear, what is the physiological relevance of these long bouts of NE release and if such prolonged bouts occur frequently? Besides, the authors need to provide direct evidence that Ca^{2+} increase in axons (seen by GCaMPs) projecting from LC can be directly co-related with the NE release from these axons (seen by dLight) (Fig. 4, Fig. 5H).

This reviewer's criticisms are very reasonable, in particular, the one about the usage of dLight to detect NA instead of DA. Indeed, other reviewers have expressed similar concerns. To this end, we re-examined extracellular NA by using the NA-optimized nLight (Patriarchi et al., 2018; collaboration with Lin Tian, UC Davis, Supplementary Fig. 3). We were pleased to see that this probe can detect lower NA levels that were not detectable by dLight. In fact, we replaced dLight measurements with nLight in Figure 2 and elsewhere. As in our answer to comment #1, we believe that the DREADD-Gi experiments provide convincing support for the detectability of physiological cAMP signals by Pink Flamindo. In the beginning of this study (Figs.1-3), we focused on exploring detailed properties of astrocytic second messengers which found astrocytic Ca^{2+} and cAMP has temporally distinct dynamics and such differences are most possibly caused by differential affinity of alpha-1 and beta receptors. However, as this reviewer mentioned, there is a concern that some of the stimuli in Figs. 1-3 might not reflect a physiological situation. Therefore, we sought to examine physiological conditions that triggers detectable cAMP elevations in Figs. 4-7. These results indicate that repeated NAergic multipeak signals induces cAMP increases in physiological conditions.

3. Authors don't provide experimental evidence that the cAMP elevations in response to either prolonged optogenetic activation of noradrenergic neurons in LC (Fig. 1) or during the fear-conditioning paradigm (Fig. 6F, H, J) are due to the direct activation of β 2-adrenergic receptors on astrocytes (SFig. 6). In the intact imaging preparations, such as one used in this study, a highly aversive stimulus like foot shock will engage almost all neurotransmitter and neuromodulator systems. Therefore, an extensive pharmacological dissection and expression analysis are essential to support a conclusion that the modest cAMP elevations seen in astrocytes are mainly due to direct activation of β 2-adrenergic receptors on these cells.

Following the suggestion of pharmacological dissection of LC-NA transmission-induced signaling in cortical astrocytes by this and Reviewer #1, we examined performed experiments using the alpha-1 antagonist prazosin, non-selective beta antagonist propranolol, beta-1 antagonist betaxolol and beta-2 antagonist ICI 118,551, which we

present in Fig. 1. We find that astrocytic Ca²⁺ was blocked by alpha-1 antagonism, while astrocytic cAMP was blocked by beta receptors, particularly the beta-1 receptor. The dominant role of the beta-1 over beta2 receptor is consistent with the Brain RNA Seq mouse transcriptome database (Zhang et al. J Neurosci 2014). We hope that the pharmacological analysis in combination with citing this well-established transcriptomic literature would suffice the concern of this reviewer. Additionally, we confirmed that NA is responsible for astrocytic Ca²⁺ and cAMP increases during fear conditioning by pharmacological dissection (prazosin + propranolol) in Supplementary Fig. 7. (lines 270-272)

4. In Figure 3, authors suggest that there is a heterogenous astrocyte population in the auditory cortex, such as cells with high Ca²⁺ and cAMP response (active cells) and cells with low Ca²⁺ and high cAMP response (less active cells), and all other possible combinations. Here, authors don't provide experimental evidence that both Ca²⁺ and cAMP sensors were co-transfected in all astrocytes analyzed. The variability in the number of "active" and "less active" cells might quite well emerge from the differences in the fraction of cells virally transfected with one or the

other sensor (Fig. 3L, M). Also, when cells were co-transfected with both Ca²⁺ and cAMP sensors, it will still be challenging to rule out the levels of expression of these sensors in each cell.

The reviewer has raised a reasonable concern that the proposed heterogeneity might have been caused by varying expression of GCaMP and Pink Flamindo. We have addressed this concern in Supplementary Fig. 4. To examine further on this issue, we plotted basal GCaMP and PinkFlamindo fluorescence signals for each astrocyte. We find that all cells that were subject to the analysis co-expressed the probes and the level of expression was highly correlated. This result supports that the observation of Ca²⁺-response-prone or cAMP-response-prone cells are not due to the biased expression of respective probes. Next, we examined if basal fluorescence intensity influences the degree of relative signal changes ($\Delta F/F$). We find that neither GCaMP nor Pink Flamindo has significant correlation between basal fluorescence level and relative signal change within the set of cells that we analyzed in this study (lines 191-200).

5. Astrocytes elevate both the second messengers, Ca²⁺ and cAMP, in response to the first foot sock, but during later foot socks these responses were diminished (Fig. 6, H). What is the mechanism of these diminished response seen in the fear conditioning paradigm, given that repeated startle can induce a robust Ca²⁺ response in astrocytes each time (Ding et al., 2013 and Paukert et al., 2014)? Also, it will be essential to see the direct dLight response, and hence NE releases, during the fear conditioning paradigm, instead of optogenetic stimulation of LC neurons shown in Fig. 7 N, O.

We would think that novelty is an important factor for the reduction of Ca²⁺ and cAMP. Startle stimulation is given abruptly, and this may be taken as “unexpected, novelty”

signal to transiently drive the noradrenergic signal. On the other hand, fear conditioning gives sound as a sign for shortly coming foot shock, therefore animals can learn and expect foot shock, therefore foot shock becomes less novel with repeated conditioning. According to this reviewer's suggestion, we measured NA level during fear conditioning with the hypothesis that NA release correlates with novelty. As a result, NA release increased in early sessions and gradually disappeared in later sessions (Supplementary Fig. 9). Together with the pharmacological blockade experiments during fear conditioning (Supplementary Fig. 7), our results suggest that released amount of NA determines the Ca^{2+} and cAMP responses after conditional stimulus (foot shock). (lines 270-272; 306-308)

6. Although the entire study was performed in the auditory cortex, authors show enhanced glycogenolysis in response to activation of cAMP pathways, using Gs activating DREADDS, in the hippocampus (SFig. 5D, E). Is this phenomenon specific for hippocampal astrocytes, and is not seen in the cortical astrocytes?

We added new cortical glycogen immunohistochemical micrographs with Gs activation by DREADD. Similar to our original observation in the hippocampus, cortical glycogen also diminished after the activation of Gs signaling pathway (Fig. 8). (lines 309-319)

F Gs-DREADDD in the cerebral cortex

Minor concerns:

- Fig. 1O – why no there no statistical significance between 1s and 2s PS, and 2s and 3s PS. Although Fig. 1L shows a clear difference during these time points.

Whiskers in a box plot is sometimes deceiving as they are often confused with error bars that represent the standard deviation or standard error of the mean. We used a standard box plot, where the length of the whisker is $1.5 \times (Q3 - Q1)$, Q3 and Q3 denote quartile (25 and 75 percentile) values.

- When comparing data, it will be helpful for the readers when the y-axis is plotted at the same scale i.e., starting from 0 (Fig. 1N-Q).

Different fluorescent probes have distinct dynamic ranges, therefore it's difficult to apply same scale.

- On page 7, 4th line (from the end), either Fig. 4L is missing, or there is a typo, and it should be Fig. 7D, L.

The typos are now corrected.

- References 35, 51 need to be edited according to the journal specifications.

The references were updated with the correct volume and page numbers.